# Structural basis of outer membrane biogenesis by the TamAB translocase

Biao Yang[1,2], Ruixin Fan[1,2], Mariana Bunoro Batista[3], Yatian Chen[1,2], Xiaofeng Duan[1,2], Rong Wang[1,2], Danyang Li[4], Phillip J. Stansfeld [3], Zhengyu Zhang [1,2] & Changjiang Dong [1,2]

The outer membrane is vital for Gram-negative bacteria, playing crucial roles in colonization, pathogenesis and drug resistance. The translocation and assembly module A and B (TamAB) nanomachinery has been reported to be involved in transport of phospholipids from the inner membrane to the outer membrane, as well as insertion of critical outer membrane proteins. However, the underlying mechanisms remain poorly understood. Here we report cryogenic electron microscopy structures of TamAB in two conformations at resolutions of 3.69 and 3.82 Å. We reveal a hybrid barrel structure formed between the first β-strand of the TamA barrel and the last β-strand of the TamB C-terminal domain, which is folded inside the β-barrel. By integrating structural analysis with functional data, biochemical assays, and molecular dynamics simulations, we identify key residues involved in TamAB interactions and characterize the mechanisms of anterograde phospholipid transport within the continuously beta-helical hydrophobic cavity of TamB. Through disulfide bond crosslinking and functional assays, we reveal that TamA crosslinks with both TamB and Ag43. Additionally, we confirm that the two cryo-EM conformational states of TamAB exist in vivo. While BAM over-expression can compensate for TamAB deletion in Ag43 insertion, it does not rescue phospholipid transport. Given that TamA and TamB orthologs are widely distributed in among bacterial and eukaryotic organisms, our findings have broad implications in cell envelope biogenesis and offer potential avenues for therapeutic development through inhibition.

Gram-negative bacteria have two membranes, designated the inner membrane (IM) and the outer membrane (OM), which are separated by a peptidoglycan cell wall within an aqueous periplasm[1]. The OM of Gram-negative bacteria is essential and plays critical roles in nutrient import, waste export, biofilm formation, colonization, drug resistance and pathogenesis[1–3], and is therefore regarded as an attractive drug target[4]. The asymmetric OM consists of four components:

lipopolysaccharide (LPS) in the outer leaflet, glycerophospholipids and lipoproteins that are predominantly in the inner leaflet, and outer membrane proteins (OMP) that span the bilayer[1,2,5]. Pathways delivering and assembling LPS, lipoproteins and OMPs to the OM have been intensively studied in recent years to reveal fundamental biological processes and mechanisms of OM biogenesis for these essential components[6–22]. LPS is transported and assembled from the IM to the

[1]Department of Thyroid and Breast Surgery, Zhongnan Hospital of Wuhan University, School of Pharmaceutical Sciences, Wuhan University, Wuhan, China. [2]Key Laboratory of Combinatorial Biosynthesis and Drug Discovery, Ministry of Education, School of Pharmaceutical Sciences, Wuhan University, Wuhan, China. [3]School of Life Sciences and Department of Chemistry, Gibbet Hill Campus, University of Warwick, Coventry, UK. [4]The Cryo-EM Center, Core facility of Wuhan University, Wuhan University, Wuhan, China. e-mail: zhengyu.zhang@whu.edu.cn; changjiangdong@whu.edu.cn

outer leaflet of the OM by a seven-subunit transenvelope super-complex, LptA-F, which consists of an ABC transporter in the IM (LptB$_2$CFG) and a 26-stranded beta-barrel and plug translocon (LptDE) in the OM. These membrane-embedded units are bridged in the periplasm by multiple copies of LptA between periplasmic domains of LptC and LptD[6-9,17,23-28]. Lipoproteins are transported to the outer membrane through the lipoprotein outer membrane localization (Lol) pathway, LolABCDE, in which LolCDE form an ABC transporter, while LolA is a periplasmic protein carrying exported lipoproteins from the IM transporter across the periplasm to the OM through LolB, itself a lipoprotein[10,11,14,29]. Nascent OMPs are synthesized in the cytoplasm and transported across the IM through SecYEG. From the periplasm, they are escorted by chaperone proteins to the OM, where the beta-barrel assembly machinery (BAM), consisting of BamA, itself an OMP, and four associated lipoproteins BamB-E, inserts nascent OMPs into the OM[4,12,13,15,16,18,19,30].

Despite these significant advancements in our understanding of bacterial membranes, the pathways for phospholipid transport between the IM and OM have remained poorly characterized until recently[2,31-33]. *Escherichia coli* membranes contain three main types of phospholipids: phosphatidylethanolamine (PE, ~75%), phosphatidylglycerol (PG, ~20%) and cardiolipin (CL, ~5%)[34]. Maintaining OM asymmetry with a certain level of phospholipids is critical for OM function and bacterial survival in different environments. Recently, several Mammalian Cell Entry (MCE) pathways and AsmA-like protein translocons were proposed to be involved in phospholipid transport between the IM and OM[35-45]. Among the MCE pathways, a phospholipid retrograde transport pathway, the Mla pathway, consisting of an ABC transporter MlaBDEFG in the IM, a periplasmic lipid transport protein MlaC, and an OM complex MlaA-OmpC/F, as well as PqiABC and LetAB pathways were reported[35,37,46]. Genetic, in vitro and in vivo functional and structural studies have been performed, which suggest that the Mla pathway transports phospholipids from the OM to the IM, while PqiABC and LetAB may transport phospholipid from the IM to OM[35,37-40,47,48]. AsmA-like proteins YhdP, TamB, and YdbH have been identified to transport phospholipid from the IM to OM, and are regarded as the long-sought-after phospholipid anterograde transporters in Gram-negative bacteria[36,41,42,49]. These three AsmA-like proteins were first predicted based on the eukaryotic AsmA-like phospholipid transporters, Atg2, Vps13 and Tic236[49], suggesting that they are evolutionarily conserved. YhdP, TamB and YdbH are reported to compensate for each other for phospholipid transport from the IM to OM in Gram-negative bacteria, with only the triple deletion being synthetically lethal. This redundancy helps to explain why these proteins were not identified earlier[36,41,42].

The translocation and assembly module A (TamA) is a protein in the OMP85 family. This integral membrane protein forms a reported complex with TamB and facilitates the insertion of certain outer membrane protein (OMP) autotransporters, which are essential for biofilm formation and colonization[50-54]. This highlights that TamAB plays an important role in both OM phospholipid and protein transport and assembly (Fig. 1a-c); however, the mechanisms by which this is achieved have yet to be elucidated[55].

Here, we present cryo-EM structures of TamAB in two distinct conformational states, alongside functional, biochemical, and biophysical assays investigating phospholipid transport and autotransporter insertion. Additionally, molecular dynamics simulations provide insights into the role of TamAB in lipid binding. Collectively, our work provides perspectives into TamAB's function in outer membrane biogenesis.

## Results

### TamAB forms a transenvelope complex
TamA consists of three N-terminal polypeptide transport-associated (POTRA) domains in the periplasm and a C-terminal 16-stranded barrel in the OM[47] (Fig. 1a-c). In contrast, TamB is proposed to be comprised of an N-terminal transmembrane helix in the IM, an extensive β-taco domain, and a C-terminal domain consisting of two helices and six β-strands[42-45] (Fig. 1a-c). TamA and TamB were co-expressed with an 8 His-tag on the C-terminal TamB (see "Methods"), and the TamA and B were co-purified by a nickel affinity column and a size-exclusion chromatography (Fig. 1d, e), suggesting that TamAB forms a stable transenvelope complex (Fig. 1c). The purified TamAB was concentrated to 3 mg/ml.

### TamB binds phospholipid
To test whether TamB binds phospholipids, the periplasmic domain of TamB (residues 43–1156) was cloned, overexpressed and purified (Supplementary Fig. 1a) (see "Methods"). Phospholipids were extracted, and thin-layer chromatography (TLC) analysis revealed that the TamB periplasmic domain binds PE and PG. In contrast, the extracts of lipoprotein chaperone LolA did not show phospholipid bands using the same method (Supplementary Fig. 1b) (see "Methods"). To further confirm whether the extracts were PE and PG, the extracts were applied on a 5800 MALDI-TOF (AB SCIEX, USA) mass spectrometer (see "Methods"), which clearly showed that the PE and PG were identified from the extracts (Supplementary Fig. 1c).

### Cryo-EM structure determination
The purified TamAB in buffer A (methods) was blotted on the Quantifoil holy carbon grid (R1.2/1.3, 300 mesh Cu). The cryo-EM samples were screened using the 200 kV Glacios, and the high-resolution data were collected on 300 kV Krios G4 (see "Methods"). 16,631 movies were used to pick the particles for the 2D classifications, and 614,471 particles were used for the four classes of 3D ab initials. After the heterogeneous and non-uniform refinements, the structure was determined to 3.58 Å (see "Methods", Supplementary Fig. 2). To check the structural heterogeneity, the 3D classification was performed to divide the particles into four groups. The TamAB hybrid barrel and non-hybrid barrel structures were resolved at a ratio around 1:1 and determined to 3.69 and 3.82 Å, respectively, based on the gold-standard Fourier shell correlation (FSC) of 0.143 (see "Methods", Supplementary Fig. 2). The high-quality density maps allowed us to build TamA residues 22-577, and TamB residues 705-1169, and TamB residues 1205-1259 for TamAB hybrid barrel structure, while TamA residues 22-577 and TamB residues 701-1169 were built for TamAB non-hybrid barrel structure (Supplementary Fig. 3a). In cases where the densities for side chains of amino acids were not evident, the side chains were not included in the TamAB models. The stoichiometric ratio of TamA to TamB was determined to be 1:1.

The TamAB complex is ~ 60 Å in length, ~ 69 Å in width and ~180 Å in height. The TamA barrel has eight extracellular loops (ECL1-8, with the longest ECL6), which seal the pore, while the seven periplasmic turns (PT1-7) extend into the periplasm (Figs. 1a and 2a–d). Overall, the two structures have a root-mean-square deviation (RMSD) of 1.9016 Å over 947 aligned Ca atoms (Fig. 3a). The key structural difference is that, in the non-hybrid barrel state, the C-terminal β-strands of TamB are not observed to engage with the TamA barrel (Fig. 2a–d).

### A hybrid barrel formed naturally between TamA and TamB
In the hybrid barrel structure, TamA is in a lateral open state, where the TamA β1 and β16 strands of the barrel are completely separated (Figs. 2b, 3a). In this configuration, TamA β1 tightly couples with TamB's last beta strand, formed by C-terminal residues 1251-1259, while the TamA β16 curves towards the inner lumen of the barrel. In this structure, nine hydrogen bonds form between the TamA β1 and the last beta strand of TamB (Fig. 3b). Moreover, ECL1 of TamA and the final β-strand hairpin loop (LP5) of TamB form three hydrogen bonds (Fig. 3b). This suggests that TamA β1 recognizes its partner protein TamB's last β-strand. In contrast, TamA β16 does not have any interaction with TamB.

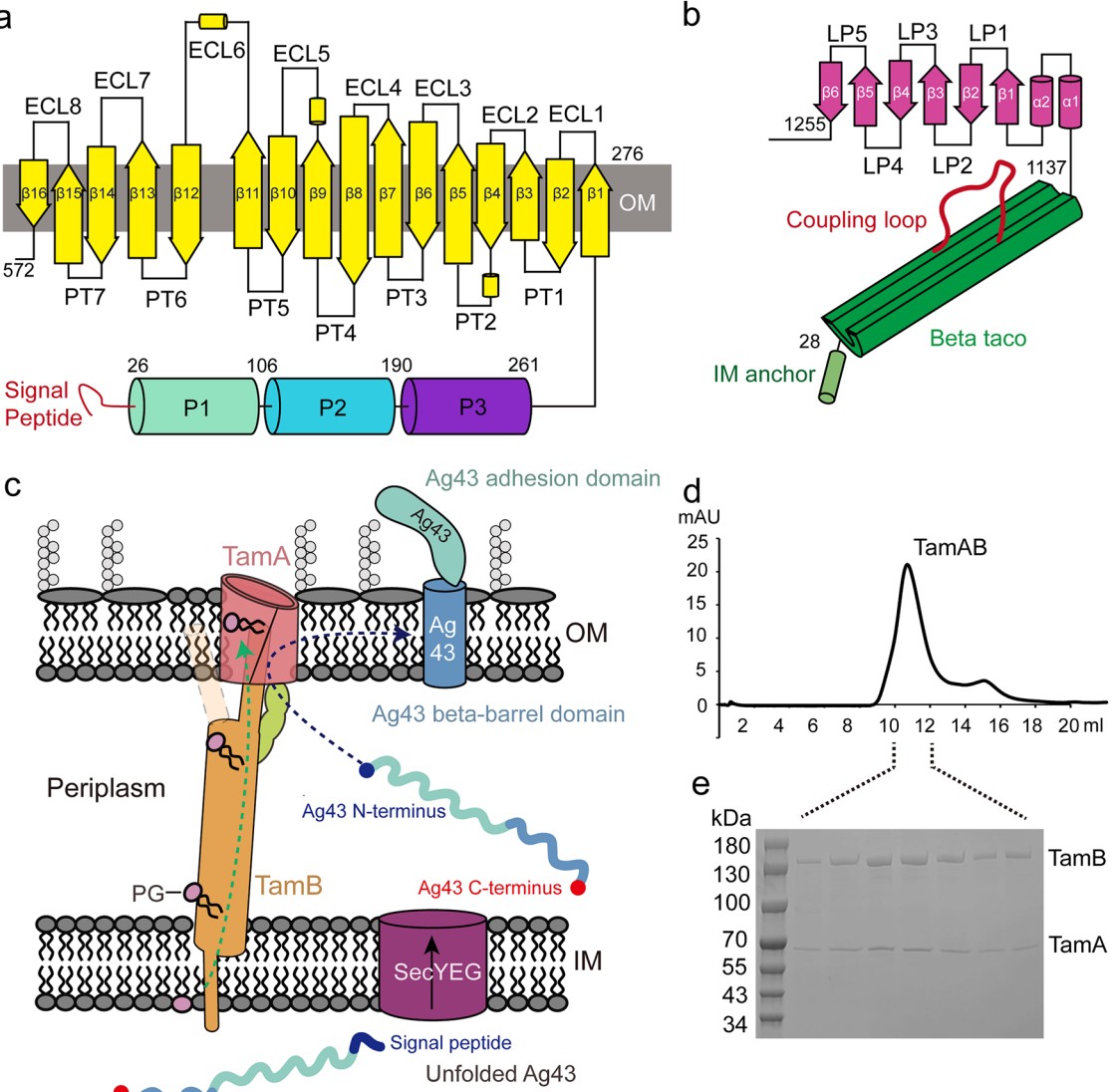

**Fig. 1 | Schematic diagram of TamAB. a** TamA contains three POTRA domains in the periplasm and a 16-stranded beta-barrel in the OM with eight extracellular loops (ECL1-8) and seven periplasmic loop (PL1-7). **b** TamB consists of an N-terminal inner membrane helix, a periplasmic beta-taco domain, and a C-terminal domain formed by two helices and six beta-strands with five loops (LP1-5). **c** Schematic representation of TamAB. TamAB forms a trans-envelope nanomachine, transporting phospholipid from the inner membrane to the outer membrane and assembling autotransporter proteins in the outer membrane. **d** Size-exclusion chromatography of purified TamAB. **e** SDS-polyacrylamide gel electrophoresis of purified TamAB. All experiments were repeated at least three times. Source data are provided as a Source Data file.

TamAB has been reported to translocate some autotransporters[50,53,56], suggesting that TamA might play a similar role as BamA in the insertion and translocation of OMPs, by recognizing the last beta strand of substrate barrels. To explore how TamA recognizes substrate OMP, we selected two well-known substrate proteins, Antigen43 (Ag43) and EhaA[50,53,56,57], as well as the partner protein TamB, and analyzed the last β-strands that potentially bind to the TamA β1. The presence of β-signal-like motifs−Φ X Φ X Φ X F, where Φ, X, and F represent hydrophobic, polar, and phenylalanine residues, respectively[13,18] (Supplementary Fig. 3d), in these OMP substrates suggests that TamA may employ a recognition mechanism similar to BamA for its substrate or partner proteins.

To test whether the interaction between the TamA β1 and TamB C-terminal β6 strand is critical, we generated the WDY strain (Δ*ydbH*, Δ*tamAB* and an *araC−arcBAD* sequence inserted in front of chromosomal *yhdP*, see "Methods"), and we demonstrated that the WDY strain is lethal in the absence of L-arabinose, consistent with previous reports[32,41,42]. We therefore used this strain to perform a functional

assay. A TamA β1 double proline mutant T270P/G273P significantly impacts cell growth (Fig. 3c), while a single mutant TamA T270A modestly slows cell growth (Fig. 3c). A deletion of residues 1245-1249 of the TamB LP5 also slows the bacterial growth (Fig. 3c), suggesting that the interaction of ECL1 of TamA and the LP5 of TamB affects TamAB function. Interestingly, BamA β1 double proline mutant and the last ECL deletion of substrate proteins were reported to be lethal or deficient in cell growth[13,16], suggesting the mechanistic conservation of TamA and BamA in OMP insertion.

## TamB C-terminal beta-strands are located inside the TamA barrel in the hybrid barrel structure

The C-terminal domain of TamB contains six β-strands and two α-helices. In our hybrid barrel structure, the four C-terminal beta-strands β3-β6 and loops LP3 and LP5 are located inside of the TamA barrel (Fig. 2a, b), while the β2 strand has weaker density and is outside the TamA barrel (Supplementary Fig. 3c). The densities for the TamB C-terminal β1 strand and the α2 helix are not visible, which indicates that they are flexible in this TamAB hybrid-barrel structure.

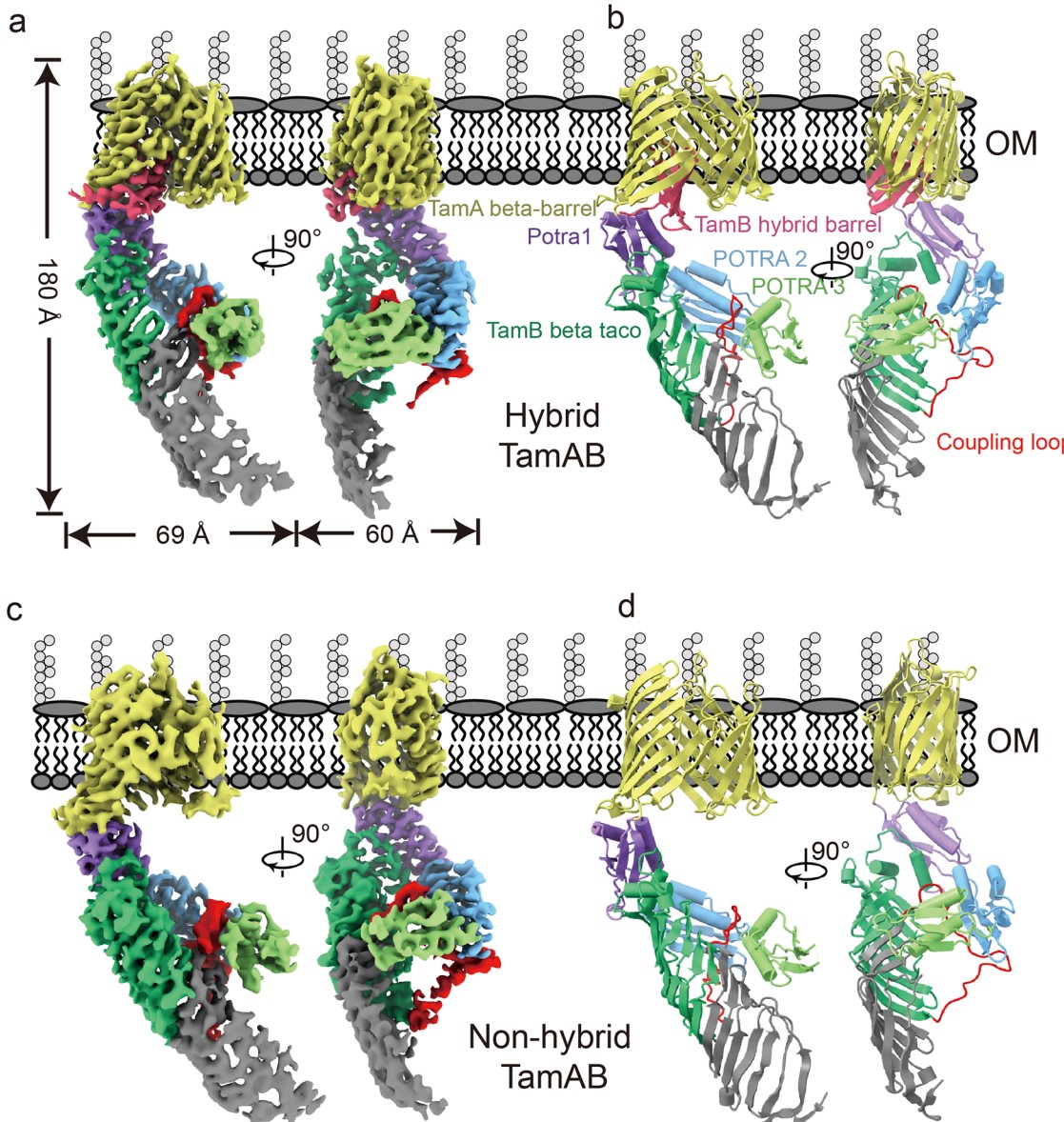

**Fig. 2 | Cryo-EM structures of TamAB in hybrid and non-hybrid barrel states. a**, Side view of the density of TamAB in the hybrid-barrel form. **b** Side view of the cartoon representation of TamAB in the hybrid-barrel form. **c** Side view of TamAB density in the non-hybrid barrel form. **d** Side view of TamAB cartoon representation of TamAB in the non-hybrid barrel form.

TamA ECL6 residue R474 forms three hydrogen bonds with the TamB LP5 (Fig. 3d), while the TamB LP3 interacts extensively with the TamA lumen residues of strands β5 and β6. These interactions are involved in hydrogen bond formation between TamA residues Y327 and TamB Y1218, TamA Q331 and TamB G1221, TamA R368 with TamB S1225 and F1223, TamA S397 and TamB V1205, as well as hydrophobic interactions between TamA K335, T348 and TamB F1223, TamA Y354, L329 and TamB V1220, TamA M391 and TamB A1227, and TamB A1243 and TamB P575 (Fig. 3d). To check the importance of the interactions between the TamB C-terminal domain and the lumen residues of TamA, the LP3 residues 1220-1227 were deleted. In our functional assay, this construct was shown to significantly impair cell growth (Fig. 3e, f).

Unlike TamAB, BAM-OMP substrate hybrid barrel (BamA-OMP substrate) structures were captured by slowing OMP folding within the BAM machinery through disulfide bond crosslinking between the OMP substrate and BAM, facilitated by mutagenesis[13,16,18,22]. These observations revealed distinct intermediates in OMP recognition,

folding, maturation, and release from the BAM machinery. Using modified EspP and BamA as OMP substrates, hybrid barrels were identified, where the BamA barrel engaged with two, three, four, or twelve EspP β-strands, as well as fourteen BamA β-strands. Notably, all β-strands of the substrate OMPs were positioned outside the BamA barrel[13,16,18,22].

A comparison between the TamAB hybrid barrel structure and BAM-OMP substrate hybrid barrel structures reveals that the TamA barrel opens wide, allowing the insertion of four TamB β-strands within its lumen (Supplementary Fig. 4).

As densities for the TamB C-terminal β1 strand (residues 1188−1196) and the α2 helix (residues 1160-1184) are not observed (Supplementary Fig. 5a, b), we sought to determine their functional roles. Functional assays following individual deletions of these segments revealed that both deletions greatly impaired cell growth. In contrast, a deletion of the TamB α1 helix (residues 1137-1154) had little to no effect (Fig. 3e and Supplementary Fig. 5).

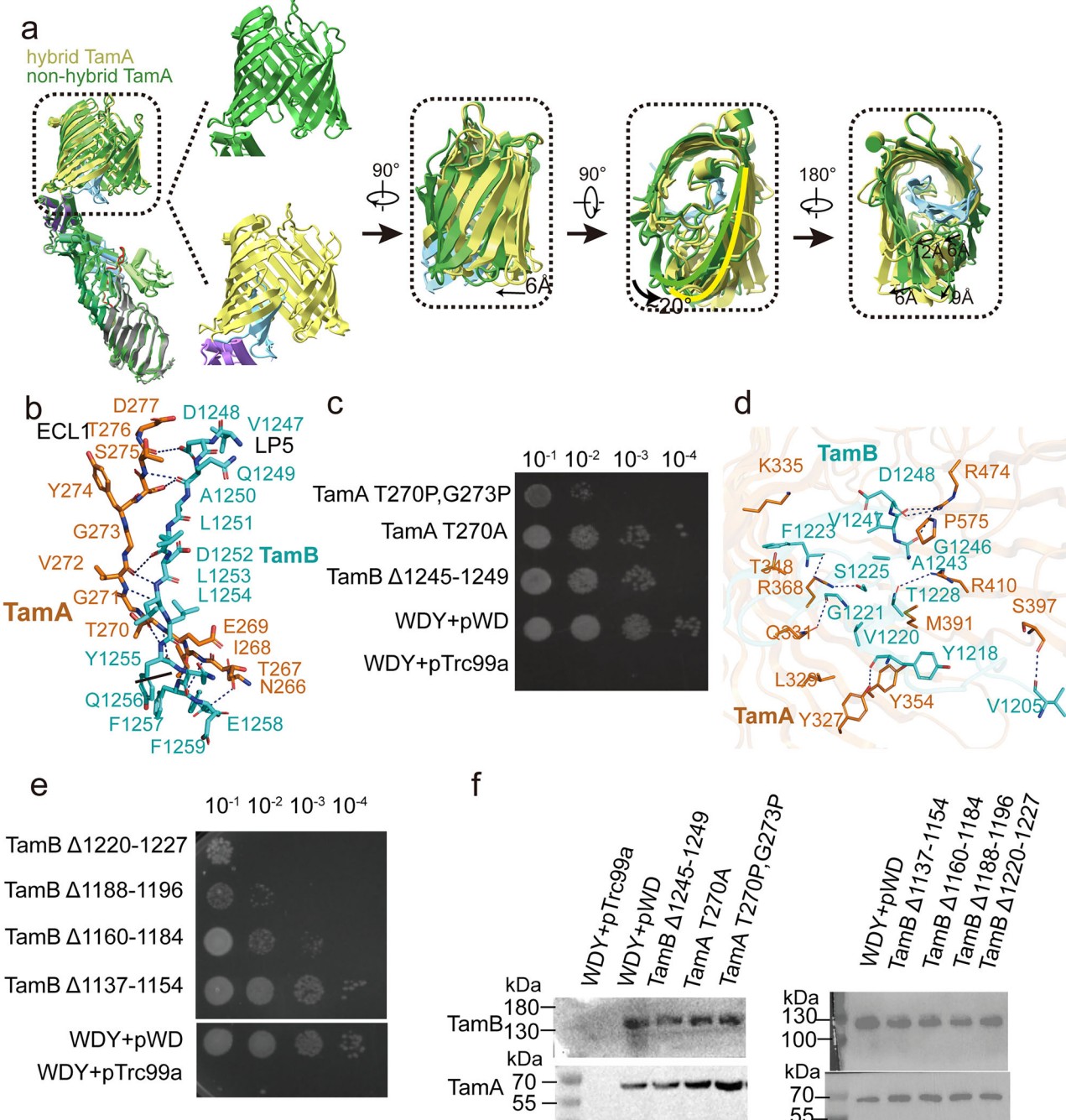

**Fig. 3 | TamA and TamB C-terminal domains form a hybrid barrel.**
**a** Superimposition of TamAB in hybrid and non-hybrid states. The main differences are that the TamB C-terminal β6 interacts with the TamA β1 and the TamB C-terminal β3-β5 fold inside the lumen of the TamA in the TamA hybrid barrel state, but no TamB C-terminal domain density is observed in the non-hybrid barrel state. To form the hybrid barrel, the TamA C-terminal beta-barrel strands turn around 20 degrees and translate about 6 Å. **b** Interactions of TamB β1 and TamB C-terminal β6,
and TamA ECL1 and TamB LP5. **c**, Functional assay of TamA β1 mutants and TamB LP5 deletion. All experiments were repeated three times. **d** The TamB C-terminal domain folds inside the lumen of TamA beta-barrel, and their interactions are illustrated. **e** Functional assays of TamB interaction residue deletions. All experiments were repeated three times. **f** Protein expression level of the TamAB mutation and deletion was determined by western blotting. All experiments were repeated three times. Source data are provided as a Source Data file.

## TamA is semi-closed at the lateral gate in the non-hybrid barrel structure

In contrast to the TamAB hybrid barrel structure, the second TamAB structure is semi-closed at the TamA lateral gate (Fig. 3a and Supplementary Fig. 6). In this state, TamA β1 and ECL1 form five hydrogen bonds with TamA β16, while the β16 residues 574-577 turn inwards, leaving the lateral gate open between TamA β1 residues I268 and G271 (Fig. 4a). TamA β1-8 and β10-β16 adopt conformational changes from

the TamAB hybrid barrel state to the TamA semi-closed second state (Fig. 3a). No densities were observed for the TamB C-terminal six β-strands and the α2 helix in this semi-closed TamAB barrel structure, suggesting that it is decoupled from TamA.

Furthermore, the TamA non-hybrid barrel structure resembles that of the crystal structure of the complete TamA (PDB code 4C00) and TamA barrel only (PDB code 4N74) with RMSD of 2.1368 Å and 1.1975 Å over 468 and 313 aligned Cα atoms (Supplementary Fig. 6a, b),

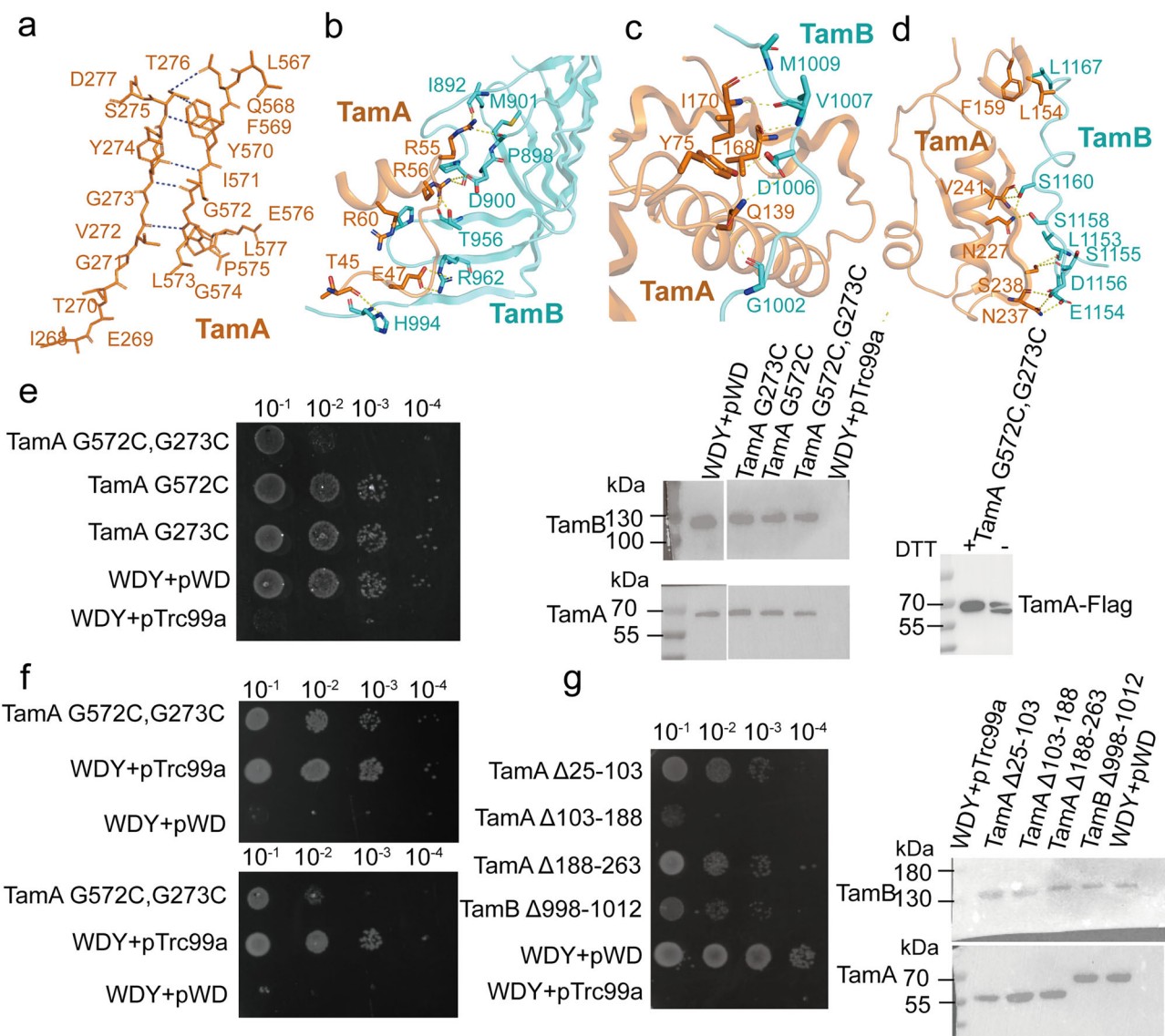

**Fig. 4 | TamA forms a half-closed barrel in the TamAB non-hybrid barrel state and TamA and TamB interactions in the periplasm. a** TamA β1 and β16 interactions in the TamAB non-hybrid barrel state. **b** TamA POTRA 1 interacts with TamB. **c** TamA POTRA 2 interacts with TamB. **d** TamA POTRA 3 interacts with the coupling loop of TamB. **e** Functional assays of the cystine mutant of TamA β1 and β16 and disulfide bond formation confirmed by western blotting. All experiments were repeated three times. **f** Functional assay of pTrc99a-TamA G572C, G273C complement to WDY on LSupplementary B agar plates with or without 1 mm TCEP. All experiments were repeated three times. **g** Functional assays of TamA POTRA 1–3 and coupling loop deletions. All experiments were repeated three times. Source data are provided as a Source Data file.

respectively. The semi-closed TamAB shares greater similarity with the TamA barrel-only crystal structure, with conformational differences in β10-13, ECL1, ECL6-8, periplasmic turns 1 and 5-7 (Supplementary Fig. 6b). The complete TamA crystal structure represents a further closed state, where TamA β1–7 strands align well with the non-hybrid barrel, but ECL1 and β8–15 adopt conformational changes to further close the barrel, with additional shifts observed in POTRA domains 1 and 2 (Supplementary Fig. 6a). A common feature among TamA non-hybrid barrel and both TamA crystal structures is their semi-closed lateral gate.

## Both hybrid and non-hybrid states exist in vivo

To confirm the in vivo existence of the TamAB hybrid-barrel state, double cysteine mutants between TamA β1 T267C/TamB β6 E1258C and TamA β1 E269C/TamB β6 Q1256C were generated, forming disulfide bonds that verified the presence of the hybrid barrel state (Supplementary Fig. 6c). To validate the in vivo existence of the TamAB

non-hybrid barrel state and assess the necessity of TamA's open and closed conformations for function, a double cysteine mutant β16 G572C/β1 G273C was generated in TamA to lock the lateral gate, with disulfide bond formation confirmed via SDS-PAGE under non-reducing conditions (Supplementary Fig. 6d). Functional assays revealed that the double cysteine mutant impairs cell growth, while the single cysteine mutants (G572C, G273C) do no. Treatment with a reducing reagent rescued the double cysteine mutant (Fig. 4e, f), confirming that both of our structural states exist in vivo.

To investigate the dynamics of both TamAB complexes, we performed atomistic molecular dynamics (MD) simulations of the TamA β-barrel (residues 276-572) in both conformations inserted into a model outer membrane. The minimum distance between the TamA β1 and TamA β16 strands was used to assess the degree of opening of the lateral gate. In the absence of TamB, the β1 and β2 strands of TamA from the hybrid state rotate inward toward the lumen of the barrel, adopting the closed conformation in 2 out of 3 simulations. Moreover,

hydrogen bonds between the TamA β1 and β16 strands are restored, stabilizing the closed conformation (Supplementary Fig. 7). No opening of the TamA beta barrel was observed for the simulations starting from the semi-closed conformation.

## TamB interacts with TamA POTRA domains

Although interactions between TamA and TamB via POTRA domains have been reported, their details remain uncharacterized[52,53]. In both TamAB structures, TamB interacts with TamA through three interfaces. POTRA domain 1 Interaction: TamA residues R55, R56, E47, R60, and T45 form hydrogen bonds, salt bridges, and hydrophobic interactions with TamB residues M901, I892, D900, P898, T956, P958, R962, and H994 (Fig. 4b). POTRA domain 2 and C-terminal coupling loop interaction: TamA residues Q139, L168, I170, and F178 (POTRA domain 2), along with A71 and Y75 (POTRA domain 1), interact with TamB C-terminal coupling loop residues V1001, G1002, V1003, D1006, V1007, M1009, L1010, and N1011 via hydrogen bonds and hydrophobic interactions (Fig. 4b, c). POTRA domain 3 Interaction: TamA residues N237, S238, N227, V241, and F159 interact with TamB residues L1153, E1154, S1155, D1156, S1158, S1160, and L1167 via hydrogen bonds and hydrophobic interactions (Fig. 4d). Extensive interactions between TamA and TamB were quantified, revealing a buried surface area of 1802.9 Å$^2$. To assess their functional significance, deletions of TamA POTRA 1 (residues 25–103), POTRA 2 (residues 103–188), POTRA 3 (residues 188–263), and the TamB C-terminal coupling loop (residues 998–1012) were generated. Functional assays demonstrated that TamA POTRA 2 deletion was lethal, TamB C-terminal coupling loop deletion significantly slowed growth, and TamA POTRA 1 or 3 deletions caused growth deficiencies (Fig. 4g). Reduced protein expression levels were observed for POTRA 3 deletion, while POTRA 1, POTRA 2, and TamB C-terminal coupling loop deletions maintained wild-type levels, indicating the critical role of TamA-TamB interactions in TamAB function.

## TamB has a curved beta-stranded structure with a hydrophobic core

In both TamAB structures, 24 β-strands could be assigned for the TamB periplasmic domains, revealing a curved, twisted architecture. The curved strands form a continuously U-shaped hydrophobic core, with hydrophilic residues on the opposite face of the β-sheet. This suggests a core binding sites for hydrophobic substrates, with exposure of the hydrophilic surface to the aqueous periplasm (Supplementary Fig. 8a). Cryo-EM densities observed above the hydrophobic core likely correspond to amphipol A8-35 molecules used during sample preparation, potentially mimicking phospholipids binding to the TamB hydrophobic core (see "Methods") (Supplementary Fig. 8b). The crystal structure of TamB residues 975-1139 (PDB code 5VTG) reveals a β-taco fold and overlaps the cryo-EM structure of the TamB hybrid barrel with an RMSD of 1.9574 Å over 131 aligned Cα atoms[58] (Supplementary Fig. 8c). There is a density in the hydrophobic cavity of the TamB crystal structure, but the density was too ambiguous to model the lauryldimethylamine N-oxide molecules used in the protein purification[58]. The TamB structure is very similar to the soluble periplasmic domains of LptC, LptA and LptD that form a continuous set of modular domains for the transport of lipopolysaccharide across the periplasm (Supplementary Fig. 9). Each of these domains has a hydrophobic core for the binding of the acyl tails of LPS.

To characterize lipid binding to TamB, coarse-grained MD simulations allowed spontaneous lipid interactions within the hydrophobic cavity. Iterative lipid addition confirmed binding of up to 15 lipids, forming a lipid-filled tunnel that spanned the cavity (Fig. 5a–e). Lipid occupancy analyses demonstrated that lipid tails bind within the hydrophobic core, while hydrophilic headgroups remain solvent-exposed (Supplementary Fig. 1a, b).

To assess lipid binding, we measured the minimum distance between the lipid and TamB, along with the lipid tail solvation number

(defined as the number of water molecules within 5 Å of the lipid acyl chains). The lipid addition procedure was repeated iteratively until 15 lipids were bound to TamB. At each step, spontaneous lipid binding to TamB was observed, as evidenced by the minimum distance between the protein and lipid (Supplementary Fig. 10a).

A reduction in lipid tail solvation numbers (Supplementary Fig. 10b) indicates that lipids associate with TamB's hydrophobic cavity rather than its external surface. The spatial lipid density map reveals that TamB accommodates multiple lipids simultaneously, forming a continuous lipid-filled tunnel that spans the length of its hydrophobic cavity (Fig. 5b). Lipid head and tail occupancy analyses demonstrate that lipid tails localize within the hydrophobic cavity of TamB's β-taco shell-shaped structure, while hydrophilic headgroups remain solvent-exposed (Fig. 5c–e)

## TamB promotes the spontaneous release of multiple lipids from bilayers

We next investigated whether lipids can spontaneously translocate from the membrane into TamB's groove. As the N-terminal region of TamB is not resolved in the cryo-EM structure, we modeled this segment (residues 0–440) using AlphaFold, and performed coarse-grained simulations with this model embedded within a lipid bilayer (Fig. 5f and Supplementary Fig. 11). The N-terminal entrance of the groove is largely composed of hydrophobic residues that sit on the membrane surface, engaging in extensive contacts with the lipid tails. In all our simulations, lipids spontaneously moved from the membrane into the hydrophobic cavity of TamB. Multiple molecules of all three lipid species present in the membrane were observed entering the cavity, diffusing along its length, and occasionally returning to the bilayer, indicating that lipid uptake by TamB is not lipid specific. (Supplementary Fig. 11). The mechanism of lipid entry into the hydrophobic cavity of TamB begins with the headgroup of the lipid moving upwards, away from the membrane surface, while the acyl tails interact with hydrophobic residues at the entrance of the groove (Fig. 5f–Step 1). The tails then penetrate fully into the cavity, with the headgroup remaining exposed to the solvent (Fig. 5f–Step 2). The lipid subsequently translocates completely into the cavity and migrates upward, allowing the headgroup to emerge into a second solvent-exposed cavity (Fig. 5f–Step 3 and 4). A second lipid initiates entry in a similar manner, facilitated by interactions between its tails and those of the preceding lipid (Fig. 5f–Step 4). Meanwhile, the first lipid continues to migrate upward along the conduit, with its headgroup re-emerging into a third cavity (Fig. 5f–Step 5 and 6). Additional lipids then enter the cavity (Fig. 5f–Step 6 and 4). These observations suggest a stepwise mechanism in which initial lipid tail–TamB's groove interactions drive lipid insertion, followed by coordinated upward migration through successive solvent-exposed cavities within TamB. This process allows multiple lipids to be accommodated simultaneously within the conduit, potentially facilitating continuous lipid transfer along its length.

## TamAB is critical for Ag43 insertion and the biogenesis pathway

Antigen 43 (Ag43) is a widely distributed autotransporter, playing a key role in bacterial aggregation and biofilm formation[53]. Ag43 consists of four domains: the signal peptide, passenger, autochaperone and translocator domains[59] (Supplementary Fig. 12). The signal peptide directs Ag43 across the inner membrane (IM) via SecYEG to the outer membrane (OM), where the translocator domain's 12 β-strands are inserted into the OM by TamAB, forming a 12-stranded β-barrel. This β-barrel exports the passenger domain across the OM to the bacterial surface, where it is cleaved and remains attached, to promote bacterial aggregation[50,52,53,56,57,60].

To assess TamAB's role in the Ag43 biogenesis pathway, bacterial aggregation assays were performed using an *E. coli* TamAB deletion strain (see "Methods"). Cell aggregation assays revealed that the TamA

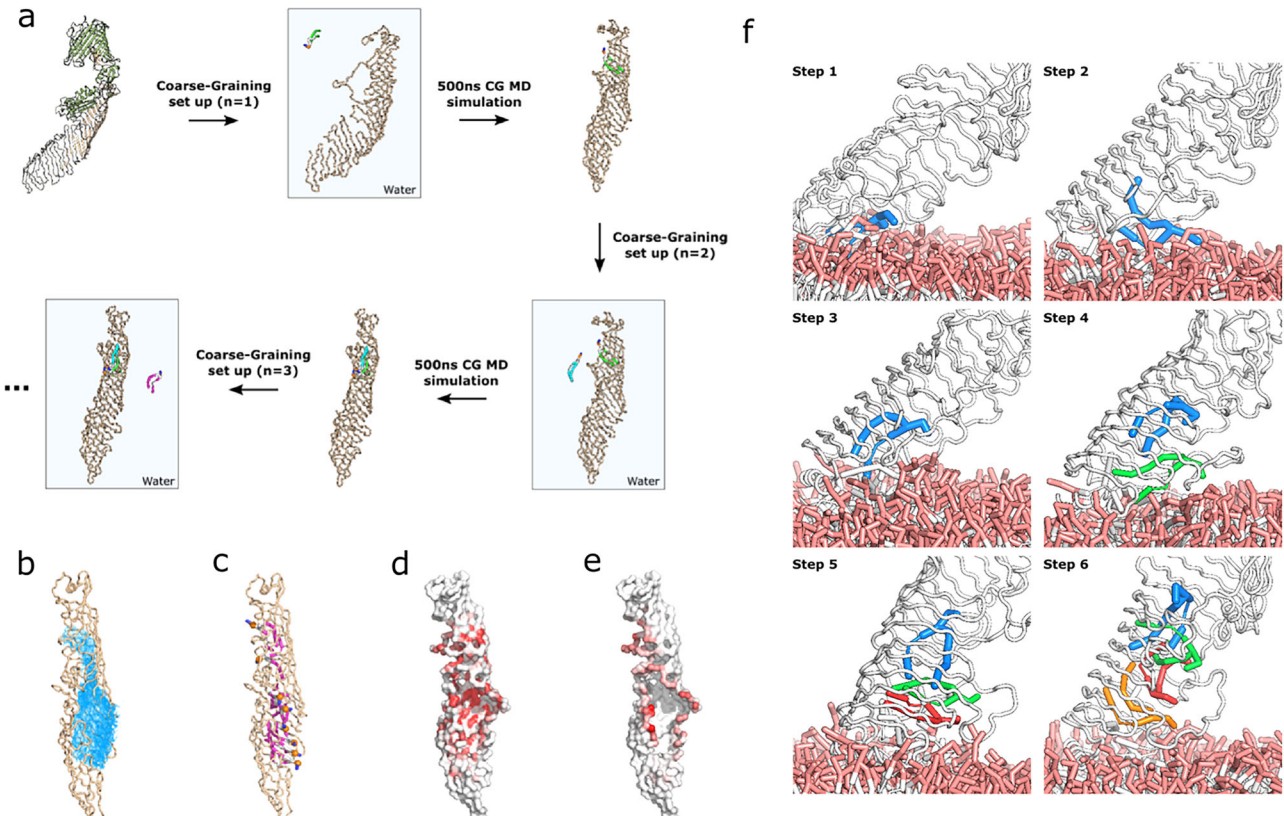

**Fig. 5 | Coarse-grained molecular dynamics simulations show lipids bind to TamB. a** CG-MD simulation protocol used to predict lipid binding. TamB atomistic structure was coarse-grained, placed in a cubic box, and one lipid was randomly placed in the bulk solvent, followed by 500 ns MD simulations. The lipid addition procedure was repeated iteratively, using the structure with n lipids bound in as the starting structure for the addition of the $(n+1)$ th lipid. **b** Spatial density maps of lipid occupancy in the last 100 ns of CG simulation for $n=7$. **c** Illustrative example of lipid binding poses showing that the lipid tails bind to the TamB hydrophobic cavity while the hydrophilic head groups are exposed to solvent. **d** Coarse-grained representation of TamB colored by the lipid tail and **e** Headgroup occupancies. **f** Sequential stages of lipid entry into the hydrophobic cavity of TamB. The process begins with upward displacement of the lipid headgroup and insertion of the acyl tails into the groove, followed by full translocation of the lipid and its migration through successive solvent-exposed cavities. Additional lipids subsequently enter the cavity, facilitated by interactions with those already accommodated within the conduit.

double proline mutant T270P/G273P and the double cystine mutant TamA G572C/G273C significantly lost protein insertion activity. Aggregation assays showed that *E. coli* cells with these TamA mutants exhibited severely reduced sedimentation activity, suggesting an improper Ag43 biogenesis pathway. Conversely, the TamA T270A mutant has little effect on TamA function, with minimal impact on bacterial aggregation, indicating a proper Ag43 biogenesis pathway (Fig. 6a).

### Both TamAB and BAM could insert Ag43 in vivo

To further explore Ag43 into the OM via TamAB, a flag tag was introduced after residue 58 of Ag43. If properly inserted and assembled in the OM, Ag43 exposure on the bacterial cell surface would allow fluorescence detection (see "Methods"). A *tamAB* deletion strain (MG1655Δ*tamAB*) was generated alongside a plasmid construct for the experiment (see "Methods"). As expected, fluorescence was strongly detected in *tamAB* deletion strains with the compensation plasmid, while TamA mutants T270P/G273P and G572C/G273C displayed weaker fluorescence. Interestingly, *tamAB* deletion strains with the empty pTrc99a plasmid also exhibited weak fluorescence (Fig. 6c). This led to the hypothesis that endogenous BAM might partially contribute to Ag43 insertion. Indeed, BAM overexpression in the *tamAB* deletion strain restored fluorescence to levels comparable to the *tamAB* compensation strains. This is consistent with sedimentation rate in the cell aggregation assays (see "Methods", Fig. 6b, c), and suggests that the endogenous BAM at its physiological expression level is partially involved in Ag43 insertion and assembly in the OM in vivo.

To obtain direct evidence that both TamAB and BAM facilitate Ag43 insertion into the OM, a slow-folding Ag43 mutation was generated by deleting Ag43 loop 1 (Gly755-Ser771) (see "Methods"). Additionally, double cysteine mutants were designed: TamA β1 residue T267C and Ag43 β12 residue Q1033C, and BamA β1 residue G431C and Ag43 β12 residue T1039C (see "Methods"). Expression of the TamAB-Ag43 double cysteine mutants confirmed the formation of disulfide bonds (Fig. 6d), supporting TamAB's role in the Ag43 OM insertion and biogenesis pathway. Similarly, the BAM-Ag43 double cysteine mutant also formed a disulfide bond (Fig. 6d), indicating BAM's involvement in Ag43 insertion and biogenesis pathway. These disulfide bonds were disrupted upon the addition of a reducing agent.

### BAM and TamAB cannot fully compensate each other

To assess functional complementarity between BAM and TamAB, we generated an arabinose-controlled BamA expression strain (see "Methods"). Complementation assays revealed that TamAB overexpression failed to rescue BamA function. Reciprocally, BAM overexpression in our WDY strain could not restore viability (Fig. 6e, f). These results demonstrate that while both complexes mediate Ag43 insertion into the outer membrane, they perform distinct, non-overlapping functions. Specifically, TamAB facilitates the folding of select outer membrane proteins (e.g., autotransporters and usher

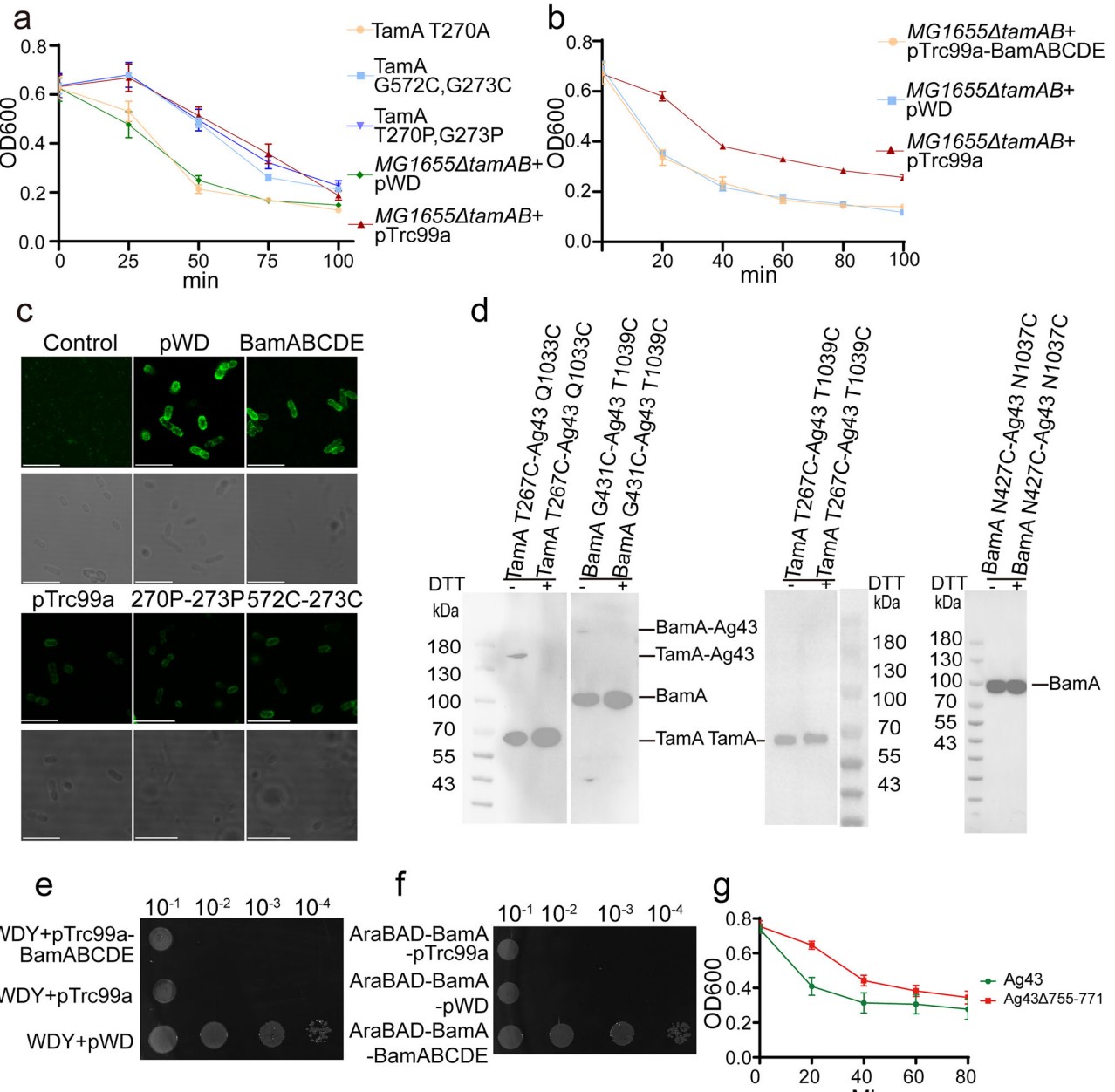

**Fig. 6 | Both TamAB and BAM could insert Ag43 in vivo. a** Sedimentation curve of *E.coli* with TamAB mutations. *E.coli* cells were left to sediment for 100 min with OD600 measurements taken at 25 min intervals. Experiments were performed in triplicate. Data are presented as mean ± SD. **b** Sedimentation curve of *E.coli* with TamAB and BamABCDE. *E.coli* cells were left to sediment for 100 min with OD600 measurements taken at 20 min intervals. Experiments were performed in triplicate. Data are presented as mean ± SD. **c**, For both the control sample (No inducer) and MG1655Δ*tamAB*-pBAD33-*ag43* with different plasmids expressing Ag43-Flag( + Ag43-Flag), a differential interference contrast image is shown in the lower frame and the corresponding fluorescence image in the upper. Scale bars are 8 μm.

**d** Analysis of crosslink mutants by western blot on a 5%DSD-PAGE gel showing the direct observation of the crosslink in BamA-Ag43 and TamA-Ag43. All experiments were repeated three times. **e** Functional assay of pTrc99a-BamABCDE complement to WDY. All experiments were repeated three times. **f** Functional assay of pTrc99a-TamAB complement to the AraBAD-BamA strain. All experiments were repeated three times. **g** Sedimentation curve of *E.coli* with Ag43 and Ag43$_{Δ755-771}$. *E.coli* cells were left to sediment for 100 min with OD600 measurements taken at 20 min intervals. All experiments were repeated three times. Data are presented as mean ± SD. Source data are provided as a Source Data file.

proteins) and mediates phospholipid transport, whereas BAM mediates the biogenesis of most outer membrane proteins.

## Discussion

Phospholipids are essential components of the Gram-negative bacterial OM, playing a critical role in maintaining its asymmetric architecture. Despite their importance, the pathways facilitating phospholipid transport from the inner membrane (IM) to the OM remain poorly understood. TamAB has been reported to be involved in both OM

phospholipid and OMP biogenesis. In this study, we present near-atomic-resolution structures of TamAB in two different states, one forming a hybrid barrel structure with TamB, the other in a semi-closed state.

Through double cysteine crosslinking, we demonstrated that the TamA β1 T267C/TamB β6 E1258C and TamA β1 E269C/TamB β6 Q1256C form the disulfide bonds (Supplementary Fig. 6c). Furthermore, we could lock the lateral gate of TamA through a disulfide bond between β1 G273C and β16 G572C (Fig. 4e, f). These experiments confirm that both structural states are possible in vivo.

In the hybrid-barrel structure, an open conformation was revealed. The first TamA β-strand binds the last TamB C-terminal beta-strand (strand 6), while TamB β-strands 5, 4 and 3, as well as LP5 and LP3, are turned inside the TamA barrel (Figs. 2a and 3d), which is distinctly different from the hybrid barrels formed by BamA and its OMP substrates (Supplementary Fig. 4).

Structural comparisons between our TamAB hybrid barrel and the BamA:RcsF structure (PDB code 6T1W) reveal significant structural differences with an RMSD of 7.362 Å over 344 Å aligned residues, when the BamA is superimposed to TamA (Supplementary Fig. 14).

To our knowledge, this is the first hybrid β-barrel structure to demonstrate that the substrate or partner protein β-strands fold within the lumen of an OMP85 family protein and the first natural hybrid barrel structure captured without mutagenesis.

Additionally, we observed that the TamB C-terminal β-strands 3−6 exhibit amphipathic properties, with hydrophilic residues facing the TamA lumen and hydrophobic residues oriented outward (Supplementary Fig. 5). This arrangement suggests that TamB β-strands are in an earlier folding state compared to previously reported BamA-substrate hybrid barrel structures[13,16,18,22].

TamB adopts an elongated taco shell-shaped structure with a hydrophobic cavity (Supplementary Fig. 8a), which is filled with lipid-like densities in our structures (Supplementary Fig. 8b). The presence of additional densities within the TamB hydrophobic cavity suggests possible phospholipid transport from the IM to the OM via TamB.

Molecular dynamics simulations show that the TamB hydrophobic groove can be continuously packed with phospholipid acyl tails, while their head groups remain solvent-exposed. Extensive interactions between TamA and TamB via POTRA domains 1, 2 and 3 indicate the formation of a tightly associated complex. Functional analysis shows that deletion of TamB's coupling loop 998-1012 and TamA POTRA domains 1 and 2 causes cell death or severe cell growth deficits (Fig. 4b−f). This suggests that the interactions are critical for TamAB function as a phospholipid transporter.

To further confirm TamB transport phospholipids, we expressed and purified the periplasmic domain of TamB. The purified TamB periplasmic domain extracts PE and PG, as determined by TLC and mass spectrometry (Supplementary Fig. 1b, 1c), suggesting that TamAB transports phospholipids. This is consistent with the report that deletion of TamAB reduces phospholipid transport from the IM to the OM[41].

Using magnetic contrast neutron reflectometry (MCNR), the TamA POTRA domain moved 33 Å when substrate Ag43 bound[54,57], suggesting that the periplasmic domain movement is important for TamAB function. A similar movement is also observed in the BAM complex[13]. We expect that the C-terminal hydrophobic cavity of TamB contacts either the C-terminal β-strands of TamB or directly with the OM to deliver the phospholipids. However, the distance between the TamB C-terminal hydrophobic cavity and the six-β-strands or OM is about 23 Å from our structures, suggesting further structural rearrangement is required. We speculate that TamA POTRA domains would carry the TamB C-terminal hydrophobic cavity to move around 23 Å to contact the OM or the C-terminal six beta strands to deliver phospholipid to the OM[44]. Genetic studies revealed that YhdP, TamB and YdbH are redundant for phospholipid transport from the inner membrane to the outer membrane. Furthermore, YhdP with P[32] radiolabeling confirmed that YhdP transports phospholipid to the outer membrane[41,42,45,61]. With structural, functional, biochemical, and molecular dynamics simulations (Figs. 2 and 5; Supplementary Figs. 1 and 8), we demonstrate that TamAB is involved in phospholipid transport.

In the TamAB semi-closed state, we propose that the TamA recognizes the β-signal motifs of certain OMPs, such as Ag43, and facilitates their insertion into the OM. Bacterial sedimentation and fluorescence assays confirm TamAB's critical role in Ag43 insertion (Fig. 6c). Disulfide bond crosslinking provides direct in vivo evidence that TamAB is involved in Ag43 insertion into the OM, while the BAM complex plays only a partial role (Fig. 6d). Although BAM over-expression can compensate for TamAB in Ag43 insertion, it does not recover the WDY strain, suggesting that the BAM does not rescue phospholipid transport. Conversely, TamAB overexpression fails to recover an AraBAD-BamA strain, indicating that TamAB likely supports a limited subset of OMPs.

Our study provides a structural and functional characterization of TamAB in two distinct states, revealing its roles in phospholipid transport and OMP insertion. By integrating biochemical assays, functional analyses, and molecular dynamics simulations, we elucidate key aspects of TamAB's function in outer membrane biogenesis. Since disrupting phospholipid transport is detrimental to bacterial survival, targeting this pathway offers promising therapeutic potential. Future studies could explore TamAB inhibition strategies as a foundation for antimicrobial development.

## Methods

### Expression and purification of TamAB

The gene fragments encoding TamAB were amplified by polymerase chain reaction (PCR) using *E. coli* K-12 genomic DNA as template. These PCR products were then assembled into a modified pTrc99a plasmid containing a C-terminal eight-histidine tag in TamB, yielding a plasmid named pTrc99a-*tamAB*(8×His). *E. coli* C43(DE3) (Weidi) harboring pTrc99a-*tamAB*(8×His) was used for protein expression.

The transformed C43(DE3) cells were cultured in Luria broth (LB) supplemented with antibiotic (ampicillin 100 μg ml⁻¹) at 37 °C until the optical density of the culture reached 0.6 at a wavelength of 600 nm (OD600). The proteins were induced by the addition of 0.1 mM iso-propyl β-d-thiogalactopyranoside (IPTG) and incubated for 16 hours at 20 °C. Cell pellets were collected and lysed in buffer A (20 mM Tris-HCl, pH 8.0, 300 mM NaCl) using a cell homogenizer at 800 bar (ATS Scientific Inc) and insoluble cell debris was removed by centrifugation at $12,000 \times g$ for 20 min at 4 °C.

The supernatant was subjected to ultracentrifugation at $140,000 \times g$ for 1 h to collect the membrane fractions. Subsequently, the cell membrane was solubilized in buffer B (20 mM Tris-Cl, pH 8.0, and 300 mM NaCl, 10 mM imidazole) supplemented with 1% (w/v) n-dodecyl-β-D-maltopyranoside (DDM) (Anatrace) at 4 °C for 1 hour. Insoluble cell debris was removed by centrifugation at $16,000 \times g$ for 30 min at 4 °C. The supernatant was collected and loaded onto a 5 ml HisTrap HP column (Cytiva) pre-equilibrated with buffer B. After washing the column with buffer C (20 mM Tris-Cl, pH 8.0, 300 mM NaCl, 60 mM imidazole) containing 0.05% DDM, the TamAB complex was eluted using buffer D (20 mM Tris-Cl, pH 8.0, 300 mM NaCl, 300 mM imidazole) containing 0.05% DDM. The eluted protein was concentrated using centrifugal filters with 100 kDa molecular weight cutoff (Merck Millipore). The protein complex was further purified by Superose 6 Increase 10/300GL (Cytiva) in buffer A containing 0.05% DDM. Peak fractions were collected, and protein fractions with the highest purity were concentrated to 3 mg ml⁻¹.

### Reconstitution of TamAB in amphipol A8-35

Purified TamAB complex in 0.05% DDM was mixed with amphipol A8-35 at a ratio of 1:3 (w/w) with gentle agitation overnight at 4 °C. The detergent was then removed using Bio-Beads SM-2 (4 °C, 3 h, 15 mg per 1 ml protein/detergent/amphipols mixture). SM-2 bio-beads were then removed by centrifugation, and the supernatant was loaded onto a Superose 6 Increase 10/300GL (Cytiva) in buffer A. The peak corresponding to TamAB was collected and validated by SDS-PAGE for further experiments. Finally, protein fractions with the highest purity were concentrated to 2.4 mg ml⁻¹.

### Site-directed mutagenesis and functional assays

An *E. coli* K12 chromosomal *tamAB* deletion strain was constructed using a modified pCas/pTargetF system as previously described[42]. The

*yhdP* gene was modified by inserting an araC-araBAD sequence amplified from pKD46 sequence into the *E. coli* K12 chromosome in front of the gene encoding *yhdP*, resulting in the araC–arcBAD-controlled *yhdP*-based *tamAB* deletion strain, The araC–arcBAD-controlled *yhdP* can only be activated by the addition of arabinose. Subsequently, a *ydbH* deletion were generated after *yhdP* modification and, finally, a Δ*tamAB* Δ*ydbH* araC–arcBAD-*yhdP* strain was constructed and named WDY (Supplementary Table. 16).

All mutations were generated by Gibson Assembly. A Flag tag (DYKDDDDK) was inserted between L26 and Q27 on TamA and a Myc tag (EQKLISEEDL) was added to the C terminal of TamB, resulting in a recombinant plasmid carrying *tamA* (Flag) *tamB* (Myc) named pWD (Supplementary Table. 16).

The gene fragments encoding Ag43 with a Flag tag after residue 58, and a Myc tag after residue 558 were amplified by PCR from *E. coli* K-12 genomic DNA and cloned into pBAD33, generating the plasmid pBAD33-*ag43* (Supplementary Table 16).

The gene fragments encoding for E. coli K12 BamABCDE were amplified by PCR and ligated into a linearized pTRC99a plasmid, resulting in a pTrc99a-BamABCDE recombinant plasmid with a Flag in BamA. Mutations of BamABCDE used in western blotting were constructed based on the pTrc99a-BamABCDE (Supplementary Table 16).

The plasmid pWD (carrying wild type or site-mutated *tamAB*) or pTrc99a-BamABCDE was transformed into the WDY strain. A single colony from each transformation was inoculated into 10 ml LB medium supplemented with 100 μg ml⁻¹ ampicillin and 0.2% L-arabinose at 37 °C overnight. Cell pellets were collected and washed three times with fresh LB medium, then diluted to an OD600 of 0.45. Tenfold serial dilution viability assays were carried out and dripped onto the LB agar plates containing ampicillin. Cell growth was recorded after an overnight culture at 37 °C. All assays were performed at least three times.

*E. coli* K12 chromosomal BamA was modified by inserting an *araC–araBAD* sequence in front of the gene encoding BamA as previously described[62] to generate the *araC–arcBAD*-controlled BamA strain named AraBAD-BamA strain. The pWD was transferred to the AraBAD-BamA strain, followed as previously described.

## Western blotting
The protein expression levels of TamA and TamB were determined by western blotting. The WDY strain with pWD, TamAB mutated pWD, or empty plasmid was inoculated into 50 ml LB supplemented with 0.2% L-arabinose, 100 μg ml⁻¹ ampicillin and 0.1 mM IPTG. After incubation at 37 °C for 6 h, the cells were harvested by centrifugation at 6000 × *g*, 4 °C for 15 min. The cell pellets were then resuspended in 0.5 ml buffer A and lysed by sonication for 5 min on ice. Insoluble cell debris was removed by centrifugation at 12,000 × *g* for 30 min at 4 °C, and the supernatant was mixed with 5× SDS-PAGE loading buffer with or without DTT.

Each 10 μl sample was loaded onto a 4–12% SDS-PAGE gel and run for 50 min. The proteins were then transferred to a PVDF membrane, washed with TBS buffer, and blocked in TBS buffer supplemented with 5% skim milk powder at 4 °C overnight. The membranes were then incubated with anti-Flag (1:1000 dilution) (Sigma, Catalog No: F1804) or anti-Myc monoclonal antibody (1:5000 dilution) (Sigma, Catalog No: 4439) at room temperature for 2 h. After incubation, the membranes were washed with TBST (20 mM Tris-HCl, pH 8, 300 mM NaCl, 0.1% Tween-20) three times and then incubated with goat anti-mouse IgG antibody (1:4000 dilution) for 1 h. After washing with TBST three times, the membranes were finally incubated with ECL substrate before imaging. The images were acquired by Monad QuickChemi. All experiments were repeated at least three times.

Ag43 loop 1 (Gly755-Ser771) was deleted based on the plasmid pBAD33-ag43, yielding pBAD33-ag43$_{\Delta755-771}$. Mutations of ag43 were constructed based on the pBAD33-ag43$_{\Delta755-771}$.

The mutant plasmids pTrc99a-BamABCDE or pWD were co-transformed with mutant plasmids pBAD33-ag43$_{\Delta755-771}$ into E. coli C43(DE3) (Weidi) for the protein expression. Detection of disulfide crosslinking in vivo by western blotting was carried out as previously described.

## Ag43 assembly assays in TamAB and BAM
The *E. coli* K12 chromosomal *tamAB* deletion strain was constructed and named MG1655Δ*tamAB* to do the autoaggregation assay[63]. pBAD33-*ag43* was co-transformed into MG1655Δ*tamAB* with plasmid pTrc99a and pTrc99a-BamABCDE and other mutations, respectively. All strains were cultured in Luria broth (LB) supplemented with antibiotics at 37 °C until reaching an optical density of 1 at 600 nm (OD600), followed by induction with 0.1% (w/v) L-(+)-arabinose at 37 °C for 2 h. 200 μl Samples were collected approximately 0.5 cm below the surface of the liquid cultures at 20-min or 25-min intervals, and OD600 was determined with a microtiter plate reader.

## Lipid extraction and TLC
A soluble TamB$_{43-1156}$ construction containing residues 43–1156 with a SUMO (Small Ubiquitin Like Modifier) tag at the N-terminal of TamB was cloned into a pTrc99a plasmid, yielding the plasmid pTrc99a-SUMO-*tamB*$_{43-1156}$ for recombinant TamB$_{43-1156}$ protein purification. The recombinant SUMO-TamB$_{43-1156}$ protein was purified to determine if TamAB could bind PL directly from the cell. The transformed BL21(DE3) cells were grown in Luria broth (LB) supplemented with 100 μg ml⁻¹ ampicillin at 37 °C until OD$_{600}$ reached 0.6. Protein expression was induced by the addition of 0.1 mM IPTG and incubated for 16 hours at 16 °C. Cell pellets were lysed in buffer A. The cells were broken using a cell homogenizer at 800 bar (ATS Scientific Inc), and insoluble cell debris was removed by centrifugation at 16,000 × *g* for 30 min at 4 °C. The supernatant was collected and loaded onto a 5 ml HisTrap HP column (Cytiva) pre-equilibrated with buffer B. The column was washed with buffer C, and the bound protein was eluted with buffer D. The eluted protein was concentrated using centrifugal filters with a 100 kDa molecular weight cutoff (Merck Millipore). Subsequently, the protein was further purified by Superdex 200 Increase 10/300 GL (Cytiva) in buffer A.

Peak fractions were collected, and protein fractions with the highest purity were concentrated to 1 mg ml⁻¹ in 2 ml mixed with 2 ml methanol and 1 ml chloroform (2:2:1, v/v/v). The samples were vortexed continuously for 5 min, incubated for 30 min at 50 °C and vortexed again for 5 min. The mixture was centrifuged (2000 × *g*, 10 min), and the lower phase was extracted and evaporated. Dried PLs were resuspended in 50 μl chloroform, and 5 μl sample was loaded onto a Silica TLC plate (Silica Gel 60 F254, Merck Millipore) and run with a 4:1:0.25 (chloroform:methanol:acetic acid) solvent according to a previously described method[64]. The TLC plate was dried for 30 min, stained with 10% (w/v) phosphomolybdic acid in ethanol and heated until PL could be visualized. The sample was then determined in negative ion reflection mode using a 5800 MALDI-TOF (AB SCIEX, USA) mass spectrometer three times. The *E. coli* polar lipids were used as lipid standards. The spectra were processed using Data Explorer Version 4.3 and presented in Excel using a scatter diagram.

## Immunofluorescence detection of Ag43
To detect the efficiency of Ag43 transfer to the outer membrane, pBAD33-*ag43* was co-transformed into MG1655Δ*tamAB* with pWD, pTrc99a, pTrc99a-BamABCDE, pTrc99a-*tamA* (T270P, G273P) *tamB* and pTrc99a-*tamA* (G572C, G273C) *tamB*, respectively. All strains were cultured in Luria broth (LB) supplemented with antibiotics at 37 °C until reaching an optical density of 1 at 600 nm (OD600), followed by induction with 5 mM L-(+)-arabinose at 20 °C for 45 min. A 1 mL of uninduced/induced culture was centrifuged at 3500 × *g* for 5 min, washed with PBS, and then applied to the poly-L-lysine-treated

microscope slides. Cells were fixed with 4% formaldehyde for 30 min at room temperature and blocked with Protein Free Rapid Blocking Buffer (Epizyme Biotech) for 1 h. Cells were incubated for 1 h at room temperature with anti-Flag (1:200 dilution) (Sigma, Catalog No: F1804). After washing three times with PBS, cells were incubated for 45 min at room temperature with Alexa Fluor 488-labeled Goat Anti-Mouse IgG at a dilution of 1:200. Followed with washing three times, Antifade Mounting Medium was added, and immunofluorescence was visualized using confocal microscopy (Leica TCS SP8 STED STED).

## Cryo-EM sample preparation and data collection
A 3 μl-drop of the purified sample at a concentration of ~2.4 mg ml⁻¹ was applied to a Quantifoil holy carbon grid (R1.2/1.3, 300 mesh Cu). Prior to sample application, all grids were glow-discharged for 50 seconds. Subsequently, the grids were frozen in liquid ethane using a Vitrobot Mark IV. The freezing process involved setting the Vitrobot Mark IV at 8 °C and 100% humidity, with no wait time, 3 s blot time, and +4 blot force. Cryo-EM images were collected on a 300 keV Titan Krios (Thermo Fisher Scientific) equipped with a K3 detector (Gatan) and a BioQuantum energy filter. Data were collected in counting mode, with 40 total frames per movie in 3 seconds, 50 electrons per Å² accumulated dose, and 0.84 Å physical pixel size. The defocus range was set between -1 to -3 μm. For more detailed information about the EM data collection parameters, please refer to Supplementary Table. 15.

## Cryo-EM structural determination
All Cryo-EM movies were processed with CryoSPARC[65]. Movies were first pre-processed by CryoSPARC live. Motion correction and contrast transfer function estimation were performed using the Patch Motion Correction and the Patch CTF estimation program. 16,631 micrographs were exported from CryoSPARC live work session into CryoSPARC workspace. 25,588,098 particles were automatically picked by Blob Picker. Several rounds of particle 2D classifications and re-extractions were performed. The particles from the good class of the 2D classifications were combined, and duplicated particles were removed, followed by a particle extraction with a box size of 360 pixels without any Fourier cropping. The selected 614,471 particles were subjected to ab initio reconstruction and heterogeneous refinement. The best three classes of 441,293 particles were combined into one class and extracted using a box size of 400 pixels for non-uniform refinement, which yielded a 3.58 Å map based on the Fourier shell correlation (FSC).

We noticed that there is ghost density near the TamA transmembrane barrel, indicating that further classifications might help to generate better maps. Then 441,293 particles were subjected to 3D classification with a mask focusing on the beta-barrel of TamA, where particles were classified into four classes. Non-uniform refinement was carried out for all four classes. Two of the four classes revealed two different maps corresponding to two distinguishing conformations of the TamAB complex. The class 3 of 110,602 particles was refined to 3.69 Å, displaying clear density for the TamAB hybrid barrel. The class 2 of 110,592 particles was refined to 3.82 Å, displaying clear density for the TamA barrel alone. Furthermore, we have determined the structures of the two other classes and found that the classes are in the hybrid-barrel and non-hybrid barrel states, the same as those of the hybrid-barrel and non-hybrid barrel structures.

## Cryo-EM model building and refinement
To build the structures of TamAB, the initial mainchain model of TamAB of the two conformations was built using the Deeptracer server[66] using the corresponding cryo-EM map as input. AlphaFold2[67] predicted models or reported crystallographic TamA models were split into multiple fragments. Each fragment was docked into the cryo-EM maps by Chimera X[68] based on the Deeptracer mainchain model and cryo-EM maps before being combined into the full models. For the hybrid TamAB, the hybrid beta-barrel adopts an unexpected structure,

and all the transmembrane beta-strands were placed into the map manually by Coot[69]. Phenix real-space refinement was used to perform the refinement for all the structural data (Supplementary Table 15). All representations of densities and models were produced with Chimera X or PyMOL.

## Coarse-grained binding simulations
To test whether TamB binds phospholipids, the periplasmic domain of TamB was extracted from the TamAB hybrid barrel structure (residues 768 to 1169) and converted to a Coarse Grained (CG) representation using Martinize2[70] and the Martini 3 force field[71]. Intra-chain elastic network with a force constant of 500 kJ mol⁻¹ nm⁻² was used to connect Cα atoms within 7 Å in the protein structure. The protein was placed in a cubic box, and one POPE molecule was randomly positioned in the bulk solvent. The system was then solvated and ionized with 0.15 M NaCl. To investigate whether TamB can bind to multiple lipids at the same time, we performed the lipid addition protocol described by Srinivasan[72]. The lipid addition procedure was repeated iteratively to avoid lipid–lipid interactions in the solvent before binding to the protein, and the structure with n lipids bound was used as the starting structure for the addition of the $(n + 1)$ th lipid (Fig. 6). This procedure was repeated until 15 lipids were bound to TamB. For each lipid addition step, 500 ns of CG MD simulation was performed with a timestep of 0.02 ps. All simulations were performed in the isothermal-isobaric ensemble at 310 K and 1 bar. Pressure was maintained at 1 bar with an isotropic compressibility of $3 \times 10^{-4}$ using the C-rescale barostat[73]. Temperature was controlled using the velocity-rescale thermostat[74], with the solvent, lipids and protein coupled to an external bath. All MD simulations were performed using GROMACS 2023[75] and analyzed using GROMACS tools and MDAnalysis[76,77]. All images were generated using PyMOL[78].

## Coarse-grained simulations of Lipid Entry into TamB
To investigate whether lipids can spontaneously enter the hydrophobic cavity of TamB from the membrane, we performed coarse-grained molecular dynamics simulations of the N-terminal region of TamB (residues 1 to 440) embedded in a POPE:POPG:CARD lipid bilayer at 7:2:1 molar ratio. The input protein was aligned according to the plane of the membrane with MEMEMBED and converted to a CG representation using the Martini 3 force field. Intra-chain elastic network with a force constant of 500 kJ mol⁻¹ nm⁻² was used to restrain the secondary structure of the protein. The system was solvated with Martini 3 waters, and NaCl was added in a concentration of 150 mM to neutralize the system. The system was energy minimized using the steepest descents method, followed by 10 ns equilibration in the NVT ensemble, then by 10 ns in the NPT ensemble, before $3 \times 5$ μs production simulations. All production simulations sampled isothermic-isobaric ensembles at 310 K using the V-rescale thermostat ($\tau_T = 1.0$), the C-rescale barostat for semi-isotropic pressure coupling at 1.0 bar ($\tau_P = 12.0$) and a time-step of 20 fs. The Reaction-Field method was used to model long-range electrostatic interactions. Bond lengths were constrained to their equilibrium values.

## TamA beta barrel all-atom MD simulations
Atomistic simulations for the open and closed conformations of TamA barrel (residues 276-572) were set up using the hybrid and non-hybrid cryo-EM structures. For the open conformation, simulations with and without TamB strands were performed. The MemProtMD pipeline[79,80] was used to run an initial 1 μs CG MD simulation to permit the equilibration of a preformed model OM bilayer around the proteins, with 100% REMP in the outer leaflet and 80% POPE and 20% POPG in the inner leaflet[81]. The input protein was aligned to the xy plane using MEMEMBED[82] and then converted to a CG representation using the Martini 2.2 force field[71]. The final snapshot was converted back to atomic details using CG2AT[83]. The complete systems were further

equilibrated for 1 ns, maintaining the structure of the protein. Three repeats of unrestrained 500 ns MD simulations were performed for each system. All simulations were performed in the isothermal-isobaric ensemble at 310 K and 1 bar using a time-step of 2 fs. Pressure was maintained at 1 bar with a semi-isotropic compressibility of $4 \times 10^{-5}$ using the Parrinello-Rahman barostat[84]. Temperature was controlled using the velocity-rescale thermostat, with the solvent, lipids and protein coupled to an external bath. The long-range electrostatic interactions were computed with the Particle Mesh Ewald method[85], while a Verlet cut-off method was used to compute the non-bonded interactions. All MD simulations were performed using GROMACS 2023[75].

## Reporting summary

Further information on research design is available in the Nature Portfolio Reporting Summary linked to this article.

## Data availability

The density maps of cryo-EM and coordinates for TamAB machinery in hybrid (open) barrel and non-hybrid (semi-closed) states were deposited in the Protein Data Bank at the access codes EMD-66762 and 9XDC, EMD-66763 and 9XDD, respectively. The Molecular Dynamics (MD) simulation parameter files are available on Figshare with the Digital Object Identifier (DOI) https://doi.org/10.6084/m9.figshare.29979559. Previously reported structures (4N74, 5VTG, 6T1W and 4C00) were used as a reference for structural comparative analysis. Source Data is provided as a Source Data file. Source data are provided with this paper.

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

## Acknowledgements

We are grateful for the excellent support from Dr. Yi Zeng, Prof. Ping Zhou and all members of the cryo-EM center and the core facility of Wuhan University. We want to express our gratitude to all our team members for their support and discussions. We thank the support from the National Key R&D Program of China (2022YFA1303500), the National Natural Science Foundation of China (Grant No: 32250710142), the Strategy and Talent Support Program of Wuhan University, and the Fundamental Research Funds for the Central Universities. PJS's lab was funded by Wellcome, MRC, BBSRC, EPSRC, NIH, JPIAMR and the Howard Dalton Center. This project made use of time on ARCHER2 granted via the UK High-End Computing Consortium for Biomolecular Simulation, HECBioSim (http://www.hecbiosim.ac.uk), supported by EPSRC (grant no. EP/R029407/1). PJS would like to thank the SCRTP at Warwick for the use of the computational infrastructure. P.J.S. acknowledges Sulis at HPC Midlands+, which was funded by the EPSRC on grant EP/T022108/1.

## Author contributions

C.D.: conceptualized, coordinated and supervised the research; R.F. and B.Y.: gene cloning, protein expression, purification, cryo-EM sample preparation; B.Y., R.F., D.L.: cryo-EM sample screening and data collection; Z.Z. and B.Y.: data processing and model building; B.Y., X.D., Y.C., and R.W.: *E. coli* TamAB deletion strain creation and functional assays. M.B., P.S.: molecular dynamics simulations; C.D., Z.Z., B.Y., M.B., and P.S.: writing the manuscript; all authors: reviewing and editing the manuscript.

## Competing interests

The authors declare no competing interests.
