## [Transparent Peer Review file · Nature Communications]

Structural basis of outer membrane biogenesis by the TamAB translocase

Corresponding Author: Professor Changjiang Dong

Version 0:

Reviewer comments:

Reviewer #1

(Remarks to the Author)

This is a revised version of the manuscript previously submitted to Nature. Although the original submission was too premature, this revised version has been substantially improved with newly obtained results that address most of the concerns I raised earlier. Therefore, I think that this work is now suitable for publication.

Reviewer #2

(Remarks to the Author)

This revision has provided a satisfactory rebuttal to my previous criticisms. I only have a few minor comments. In the abstract, on line 21, please change it to read "However, the underlying mechanisms ...". Also in the abstract, on line 41, remove "in". Finally, the title to the Supplementary Table 1 should read "Bacterial strains and plasmids used in this study."

Reviewer #3

(Remarks to the Author)

The TAM (translocation and assembly module) complex is implicated in both lipid trafficking and the biogenesis of outer membrane proteins (OMPs), including autotransporters and usher proteins. While structural data for TamA and a portion of TamB have previously been reported, the precise nature of their interaction and the mechanism by which they execute their functions remain unclear. In this study, the authors present the first cryo-EM structure of TamA in complex with full-length TamB, captured in two distinct conformational states—one featuring a hybrid β -barrel formed between TamA and TamB, and the other lacking this interaction. To further probe these conformations, the authors conducted mutational analyses along key interaction interfaces and employed functional assays to evaluate their effects. These findings provide critical structural insights into the TAM complex and lay the groundwork for future mechanistic studies aimed at elucidating its dual role in lipid transport and OMP biogenesis. Although the structural characterisation of both the hybrid and non-hybrid forms represents a significant advance, the manuscript provides limited experimental validation of the underlying mechanisms governing lipid trafficking or OMP assembly, which tempers the overall enthusiasm.

Primary Comments:

1. Dual structures not fully addressed: The authors do not offer a hypothesis for the presence of two distinct structural states or their potential functional significance. If the overall architecture is otherwise similar, why does a hybrid barrel form in only one of them?
2. Unexplored 3D classes: The manuscript focuses on two primary 3D classes, yet other seemingly well-resolved 3D classes are neither analysed nor discussed. These classes appear to have sufficient resolution to identify the presence or absence of a hybrid barrel or reveal alternative conformations. Were these additional classes investigated further?
3. Valine-to-aspartate mutation (Line 204): The V→D mutation represents a major physicochemical change, likely affecting more than just hydrophobic interactions. Surface presentation of this mutant should be experimentally verified, and a more comprehensive mutational analysis is necessary. Can the authors demonstrate a direct interaction in the wild type that is disrupted in the mutant? Was this mutant evaluated using Ag43 aggregation assays?

4. Lipid binding at $\beta 1$ in TamA: In previous TamA crystal structures, a lipid was observed bound near the exposed edge of $\beta 1$ in the barrel domain. Is there any evidence of similar lipid binding in the non-hybrid structure?
5. Crosslinking efficiency: Mutants show ~50% crosslinking efficiency. Would this level not be sufficient to support at least partial functionality in vivo, such as maintaining cell growth?
6. Lack of experimental lipid binding validation: While molecular dynamics (MD) simulations suggest a lipid-binding propensity, no experimental evidence for lipid binding is provided. Techniques like isothermal titration calorimetry (ITC) could help verify binding, determine lipid-to-protein ratios, and align computational predictions with experimental data. Identification of specific lipid species involved would further support the proposed role in lipid trafficking.
7. Mechanistic gaps: The manuscript lacks details on the mechanism of lipid transport—how lipids are loaded into TamB, translocated along its length, or delivered to the outer membrane. Similarly, there is no mechanistic insight into how TAM facilitates OMP biogenesis. A proposed mechanistic model outlining how TamAB coordinates lipid transport and OMP assembly would enhance the manuscript. Can both functions occur simultaneously, or are they mutually exclusive?
8. BAM compensation claim: On page 15, the authors state that BAM is only “partially” involved in Ag43 biogenesis in the absence of TamAB. However, the data suggest that BAM may fully compensate. Can this claim be clarified or better supported?
9. Ag43 loop1 mutant (G755–S771): This mutation is not referenced in prior literature and is insufficiently characterised in the current study. Either cite previous work showing this mutant exhibits slow folding or provide a more detailed characterisation here.
10. Underdeveloped final section: The final section of the Results is brief and lacks clarity. The authors suggest that BAM cannot replace the lipid trafficking role of TamAB, while TamAB cannot fulfil BAM's broader role in OMP biogenesis. This point requires a more detailed explanation and supporting data.
11. Weak crosslinking and potential artefacts: Crosslinking signals for Ag43 interactions with TamA and BamA are faint, raising concerns about efficiency. Negative controls at non-interacting sites should be included. The TamA E269C mutant shows reduced expression—possibly indicating instability—yet exhibits crosslinking levels comparable to other mutants. Do the authors have an explanation for this discrepancy?

Secondary Comments:

12. The manuscript would benefit from thorough proofreading to correct numerous typographical, grammatical, and formatting errors, including paragraph indentation and capitalisation issues.
13. In Figure 1e, TamA appears to be present in excess. Was TamA observed alone in the cryo-EM datasets?

Version 1:

Reviewer comments:

Reviewer #1

(Remarks to the Author)

This is a second revision of the previously revised manuscript. I carefully reviewed the manuscript along with the reviewers' comments and the authors' responses, and found that the authors have addressed Reviewer 3's concerns appropriately. The manuscript is now scientifically inspiring and will deserve publication. Although direct experimental evidence for lipid transport is still lacking, such direct demonstrations are generally difficult to achieve. Therefore, the combination of lipid binding observations and MD simulations is a commonly accepted approach in lipid transport research.

Reviewer #3

(Remarks to the Author)

Thank you to the authors for their attention to the critiques and their considered responses. The authors have satisfactorily addressed all of this reviewer's major concerns. I agree that some of the comments fall beyond the scope of the current study and can reasonably be explored in future work.

Just a few points of clarification:

1. It would be helpful to state explicitly whether TAM and BAM can cooperate in the folding of OMPs, or whether each OMP is folded exclusively by either TAM or BAM. Additionally, how might this process be regulated — is it a first-come, first-served mechanism, or an on/off pathway selection for each?
2. Regarding the aggregation assay of Ag43, it would be useful to distinguish more clearly between 'biogenesis' and 'folding'. It remains possible that the mutation impacts a specific step in the biogenesis pathway without necessarily affecting folding or membrane insertion. This point does not yet appear to have been fully addressed.

Reviewer #1 (Remarks to the Author):

This is a revised version of the manuscript previously submitted to Nature. Although the original submission was too premature, this revised version has been substantially improved with newly obtained results that address most of the concerns I raised earlier. Therefore, I think that this work is now suitable for publication.

Thank you very much for your support!

Reviewer #2 (Remarks to the Author):

This revision has provided a satisfactory rebuttal to my previous criticisms. I only have a few minor comments. In the abstract, on line 21, please change it to read "However, the underlying mechanisms ..." Also in the abstract, on line 41, remove "in". Finally, the title to the Supplementary Table 1 should read "Bacterial strains and plasmids used in this study."

Thank you very much for the suggestions! We have revised the manuscript accordingly (Line 29. Supplementary Table 2).

Reviewer #3 (Remarks to the Author):

The TAM (translocation and assembly module) complex is implicated in both lipid trafficking and the biogenesis of outer membrane proteins (OMPs), including autotransporters and usher proteins. While structural data for TamA and a portion of TamB have previously been reported, the precise nature of their interaction and the mechanism by which they execute their functions remain unclear. In this study, the authors present the first cryo-EM structure of TamA in complex with full-length TamB, captured in two distinct conformational states—one featuring a hybrid β -barrel formed between TamA and TamB, and the other lacking this interaction. To further probe these conformations, the authors conducted mutational analyses along key interaction interfaces and employed functional assays to evaluate their effects. These findings provide critical structural insights into the TAM complex and lay the groundwork for future mechanistic studies aimed at elucidating its dual role in lipid transport and OMP biogenesis. Although the structural characterisation of both the hybrid and non-hybrid forms represents a significant advance, the manuscript provides limited experimental validation of the underlying mechanisms governing lipid trafficking or OMP assembly, which tempers the overall enthusiasm.

Primary Comments:

1. Dual structures not fully addressed: The authors do not offer a hypothesis for the presence of two distinct structural states or their potential functional significance. If the

overall architecture is otherwise similar, why does a hybrid barrel form in only one of them?

Thank you for your comment. TamAB has two functions: phospholipid transport and some outer membrane assembly. Our work demonstrates that TamAB exists in two structural states in vivo—the hybrid barrel state and the non-hybrid barrel state. We hypothesize that the hybrid barrel structure facilitates phospholipid transport, while the non-hybrid barrel structure mediates some outer membrane protein assembly.

2. Unexplored 3D classes: The manuscript focuses on two primary 3D classes, yet other seemingly well-resolved 3D classes are neither analysed nor discussed. These classes appear to have sufficient resolution to identify the presence or absence of a hybrid barrel or reveal alternative conformations. Were these additional classes investigated further?

Thank you for your comments! Among the other two classes, class 0 adopts a hybrid barrel conformation (4.06 Å resolution), while class 3 exhibits a non-hybrid barrel conformation (4.25 Å resolution). The 4.06 Å hybrid barrel structure is superimposable with the 3.69 Å TamAB hybrid barrel structure we reported in the manuscript. Similarly, the 4.25 Å non-hybrid barrel structure closely resembles the 3.82 Å TamAB non-hybrid barrel structure from our study. Given these similarities, we chose not to include these data in the manuscript.

3. Valine-to-aspartate mutation (Line 204): The V→D mutation represents a major physicochemical change, likely affecting more than just hydrophobic interactions. Surface presentation of this mutant should be experimentally verified, and a more comprehensive mutational analysis is necessary. Can the authors demonstrate a direct interaction in the wild type that is disrupted in the mutant? Was this mutant evaluated using Ag43 aggregation assays?

Thank you for your comment! We generated both V→E and V→A mutations and found that the V→E mutation led to cell growth deficiency, whereas the V→A mutation had no detectable effect (Fig. 3e, Supplementary Fig. 6d). However, we believe a more comprehensive analysis of these findings would be valuable in future studies. As such, we have decided to remove the V548 mutagenesis results from the current manuscript.

4. Lipid binding at $\beta 1$ in TamA: In previous TamA crystal structures, a lipid was observed bound near the exposed edge of $\beta 1$ in the barrel domain. Is there any evidence of similar lipid binding in the non-hybrid structure?

Thank you for your comment! Upon re-examining the non-hybrid structure, we observed that the TamA transmembrane barrel - including the $\beta 1$ edge is fully surrounded in A8-35 micelles. However, we did not detect clear density corresponding to a specifically bound lipid at the exposed $\beta 1$ edge of the barrel domain.

5. Crosslinking efficiency: Mutants show ~50% crosslinking efficiency. Would this level not be sufficient to support at least partial functionality in vivo, such as maintaining cell growth?

Thank you for your comments. Our structural studies reveal that the hybrid-barrel and non-hybrid barrel conformations exist in an approximately 1:1 ratio. Furthermore, our data demonstrate that disulfide crosslinking significantly compromises cell growth. We speculate that this crosslinking may have additional, as yet uncharacterized effects on bacterial physiology.

6. Lack of experimental lipid binding validation: While molecular dynamics (MD) simulations suggest a lipid-binding propensity, no experimental evidence for lipid binding is provided. Techniques like isothermal titration calorimetry (ITC) could help verify binding, determine lipid-to-protein ratios, and align computational predictions with experimental data. Identification of specific lipid species involved would further support the proposed role in lipid trafficking.

We appreciate your comments. Our data demonstrate that the TamB periplasmic domain binds phospholipids, as shown by thin-layer chromatography (TLC), and mass spectrometry confirmed the identity of the extracted material (Supplementary Fig. 1). Furthermore, our TamAB structural data reveal extra densities in the TamB periplasmic domain's groove that might be A8-35 or endogenous lipids. However, due to the inherent experimental difficulties of working with the TamAB membrane protein complex, determining precise phospholipid binding affinities and lipid-to-protein ratios using isothermal titration calorimetry (ITC) has proven technically challenging. This experiment is beyond the scope of this manuscript and will be explored in future studies.

Fig. S1. | TamB periplasmic domain binds phospholipid. **a**, TamB periplasmic domain purification by size-exclusion chromatography. **b**, Phospholipids were extracted from the purified TamB periplasmic domain and detected by thin-layer chromatography. The PE, PG and *E. coli* polar lipids were used as lipid standards. **c**, Phospholipids were extracted from the purified TamB periplasmic domain and detected by 5800 MALDITOF (AB SCIEX, USA) mass spectrometer. The *E. coli* polar lipids were used as lipid standards.

7. Mechanistic gaps: The manuscript lacks details on the mechanism of lipid transport—how lipids are loaded into TamB, translocated along its length, or delivered to the outer membrane. Similarly, there is no mechanistic insight into how TAM facilitates OMP biogenesis. A proposed mechanistic model outlining how TamAB coordinates lipid transport and OMP assembly would enhance the manuscript. Can both functions occur

simultaneously, or are they mutually exclusive?

We appreciate your comments. To investigate phospholipid transport from the inner membrane to TamAB, we conducted additional coarse-grained molecular dynamics simulations of the N-terminal segment of TamB embedded in a lipid bilayer.

Our MD simulations indicate that phospholipids spontaneously move from the membrane into TamB hydrophobic groove. Moreover, multiple lipids can occupy the groove simultaneously, suggesting a step-wise, continuous transport mechanism. We have updated the manuscript to include this data (Fig. 5).

Regarding subsequent transfer to the outer membrane, we also performed CG-MD simulations of the C-terminal region of TamB in complex with TamA, based on the hybrid TamAB cryo-EM structure. Lipids were placed inside TamB's groove to assess whether transfer to the outer membrane could occur. We did not observe any lipid dissociation events from the groove to the outer membrane (**Review response Figure 1**). We, therefore, propose that TamAB undergoes a conformational change to establish connectivity between its groove and the outer membrane, thereby facilitating phospholipid delivery.

For TAM-mediated OMP biogenesis, our work suggests that TamA may employ a mechanism analogous to BamA for certain outer membrane insertion events. Specifically, we generated a slow-folding Ag43 variant (Δ Gly755-Ser771) and introduced double cysteine mutants between TamA β 1 and Ag43 β 12. Crosslinking analysis revealed disulfide bond formation between TamA's first β -strand and Ag43's terminal β -strand, supporting mechanistic parallels between TamAB and the BAM complex in OMP assembly.

We hypothesize that the hybrid-barrel TamAB conformation mediates phospholipid transport, while the non-hybrid-barrel state facilitates some OMP assembly. These structurally distinct states coexist *in vivo*, suggesting TamAB may simultaneously perform both functions. Our findings establish a framework for future mechanistic studies of TamAB's dual roles in outer membrane biogenesis.

Review response Figure 1. Initial and final frames of a coarse-grained simulation of the hybrid TamAB complex with POPE molecules placed inside the TamB cavity. Throughout the simulations, no lipid dissociation events from TamB to the outer membrane were

observed.

Fig. 5 | Coarse-grained molecular dynamics simulations show lipids bind to TamB. a, CG-MD simulation protocol used to predict lipid binding. TamB atomistic structure was coarse-grained, placed in a cubic box and one lipid was randomly placed in the bulk solvent, followed by 500ns MD simulations. The lipid addition procedure was repeated iteratively, using the structure with n lipids bound in as the starting structure for the addition of the $(n+1)$ th lipid. **b**, Spatial density maps of lipid occupancy in the last 100 ns of CG simulation for $n=7$. **c**, Illustrative example of lipid binding poses showing that the lipid tails bind to TamB hydrophobic cavity while the hydrophilic head groups are exposed to solvent. **d**, Coarse-grained representation of TamB coloured by the lipid tail and **e**, Headgroup occupancies. **f**, Sequential stages of lipid entry into the hydrophobic cavity of TamB. The process begins with upward displacement of the lipid headgroup and insertion of the acyl tails into the groove, followed by full translocation of the lipid and its migration through successive solvent-exposed cavities. Additional lipids subsequently enter the cavity, facilitated by interactions with those already accommodated within the conduit.

8. BAM compensation claim: On page 15, the authors state that BAM is only “partially”

involved in Ag43 biogenesis in the absence of TamAB. However, the data suggest that BAM may fully compensate. Can this claim be clarified or better supported?

We thank you for your comments. To investigate TamAB's role in Ag43 folding, we generated a $\Delta tamAB$ strain and performed fluorescence assays. The observed weak fluorescence signal indicated residual Ag43 folding activity. We hypothesize that this partial folding may be mediated by endogenous BAM complex operating at its physiological (low) expression level.

To test this hypothesis, we overexpressed the BAM complex in the $\Delta tamAB$ strain and found that it fully restored Ag43 folding, compensating for TamAB deletion.

To demonstrate dual involvement of both TamAB and BAM complex in Ag43 folding, we employed a slow-folding Ag43 variant and generated double cysteine mutants between: TamA $\beta 1$ (or BamA $\beta 1$) and Ag43 $\beta 12$. Crosslinking analysis successfully captured both TamA-Ag43 and BamA-Ag43 complexes, providing direct evidence that both systems participate in Ag43 folding.

9. Ag43 loop1 mutant (G755–S771): This mutation is not referenced in prior literature and is insufficiently characterised in the current study. Either cite previous work showing this mutant exhibits slow folding or provide a more detailed characterisation here.

Thank you for your comments. Our additional Ag43 aggregation assays demonstrated that the loop1 deletion mutant exhibits slower aggregation kinetics compared to wild-type Ag43 (Fig. 6g). This observation validates our experimental approach, demonstrating that the $\Delta G755-S771$ deletion slows Ag43 folding kinetics.

Fig. 6g. The Ag43 loop 1 deletion (G755-S771) slows the Ag43 folding. We used the bacterial aggregation to determine the Ag43 folding. The Ag43 loop 1 mutant slows the bacterial aggregation.

10. Underdeveloped final section: The final section of the Results is brief and lacks clarity. The authors suggest that BAM cannot replace the lipid trafficking role of TamAB, while TamAB cannot fulfil BAM's broader role in OMP biogenesis. This point requires a more detailed explanation and supporting data.

Thanks for the comments. Our work demonstrated that both TamAB and the BAM complex participate in Ag43 folding. To investigate whether these systems are

functionally interchangeable, we first tested if TamAB could complement an arabinose-controlled BamA strain. TamAB failed to rescue this strain, indicating that while TamAB can fold certain OMPs like Ag43, it cannot replace the BAM complex for other OMP folding.

Furthermore, while we showed that BAM overexpression fully compensates for Ag43 folding in Δ tamAB strains, we questioned whether BAM could replace TamAB's complete functionality. Using our WDY strain (Δ ydbH, Δ tamAB with araC-araBAD inserted before chromosomal yhdP), we found that BAM overexpression failed to rescue this strain. This demonstrates that although over-expressed BAM can assume TamAB's role in Ag43 folding, it cannot mediate phospholipid transport- a key function of the TamAB complex. We have revised the manuscript accordingly.

11. Weak crosslinking and potential artefacts: Crosslinking signals for Ag43 interactions with TamA and BamA are faint, raising concerns about efficiency. Negative controls at non-interacting sites should be included. The TamA E269C mutant shows reduced expression—possibly indicating instability—yet exhibits crosslinking levels comparable to other mutants. Do the authors have an explanation for this discrepancy?

Thanks for your comments. Our additional negative controls using double cysteine mutants (TamA β 1-T267C/Ag43 β 12-T1039C; BamA N427C-Ag43 N1037C) confirmed the absence of disulfide bond formation (Fig. 6d).

Regarding the TamA-E269C mutant: While this variant showed lower level of the protein, we observed corresponding decreases in crosslinking efficiency. This suggests that despite lower level of the protein, the mutant TamAB complex retains the ability to recognize Ag43's terminal β -strand and form transient interactions. However, to prevent potential misinterpretation of these results, we have elected to exclude this dataset from the current manuscript.

Fig. 6d. TamA and BamA recognition of Ag43 for Ag43 folding. Double cysteine mutants, TamA T267C-Ag43 (loop 1 deletion) Q1033C, and BamA G431C-Ag43 (loop 1 deletion) T1039C could form disulfide bonds, while two controls did not observe the disulfide bond formation.

Secondary Comments:

12. The manuscript would benefit from thorough proofreading to correct numerous typographical, grammatical, and formatting errors, including paragraph indentation and capitalisation issues.

Thank you for the comments. We have revised the entire manuscript accordingly.

13. In Figure 1e, TamA appears to be present in excess. Was TamA observed alone in the cryo-EM datasets

Thank you for your observation. After thorough processing of our cryo-EM data, we found no evidence of particles consisting solely of TamA in our datasets.

REVIEWERS' COMMENTS

Reviewer #1 (Remarks to the Author):

This is a second revision of the previously revised manuscript. I carefully reviewed the manuscript along with the reviewers' comments and the authors' responses, and found that the authors have addressed Reviewer 3's concerns appropriately. The manuscript is now scientifically inspiring and will deserve publication. Although direct experimental evidence for lipid transport is still lacking, such direct demonstrations are generally difficult to achieve. Therefore, the combination of lipid binding observations and MD simulations is a commonly accepted approach in lipid transport research.

Thank you very much for the support!

Reviewer #3 (Remarks to the Author):

Thank you to the authors for their attention to the critiques and their considered responses. The authors have satisfactorily addressed all of this reviewer's major concerns. I agree that some of the comments fall beyond the scope of the current study and can reasonably be explored in future work.

Thank you very much for the comments!

Just a few points of clarification:

1. It would be helpful to state explicitly whether TAM and BAM can cooperate in the folding of OMPs, or whether each OMP is folded exclusively by either TAM or BAM. Additionally, how might this process be regulated — is it a first-come, first-served mechanism, or an on/off pathway selection for each?

Thank you for the comment. We propose that while the BAM complex is responsible for folding the majority of outer membrane proteins (OMPs), it folds specific OMPs, autotransporters like Ag43 with low efficiency. The TAM complex acts as a specialized facilitator for such OMPs. This is supported by our observation that a Δ TamAB strain exhibits reduced Ag43 fluorescence, indicating that only a fraction of Ag43 is folded by the endogenous BAM complex alone. However, overexpressing BAM in this deletion background restores full fluorescence, demonstrating BAM's capacity to fold Ag43 in the absence of TAM in vivo. Consequently, we conclude that OMP allocation is not a first-come, first-served process, but rather involves selective recognition of substrates by the BAM and TAM complexes.

2. Regarding the aggregation assay of Ag43, it would be useful to distinguish more clearly between 'biogenesis' and 'folding'. It remains possible that the mutation impacts a specific step in the biogenesis pathway without necessarily affecting folding or membrane insertion. This point does not yet appear to have been fully addressed.

Thank you for this comment. We agree that the observation of aggregation in the Ag43 assay does not necessarily indicate a protein folding defect *per se*; it could instead result from the disruption of a specific step in the Ag43 biogenesis pathway. Therefore, concluding that a mutation 'affects folding' based solely on an aggregation assay may not accurately reflect the underlying mechanism. The more precise interpretation is that the mutation 'disrupts biogenesis,' and further experiments are required to determine if the defect lies in the folding step itself or an upstream event in the assembly pathway. We have revised the manuscript accordingly to reflect this distinction in line 337, line 346, line 351-353.